# GTPase-activating protein DLC1 spatio-temporally regulates Rho signaling

Lucien Hinderling[1,2†], Max Heydasch[1†], Giliane Rochat[1], Laurent Dubied[1], Jakobus van Unen[1], Maciej Dobrzynski[1], Olivier Pertz[1]*

[1]Institute of Cell Biology, University of Bern, Bern, Switzerland; [2]Graduate School for Cellular and Biomedical Sciences, University of Bern, Bern, Switzerland

## eLife Assessment

This study presents a **valuable** finding on how the GAP DLC1, a deactivator of the small GTPase RhoA, regulates RhoA activity globally as well as at Focal Adhesions. Using a new acute optogenetic system coupled to a RhoA activity biosensor, the authors present **convincing** evidence that DLC1 amplifies local Rho activity at Focal Adhesions. Thanks to modeling, they show that DLC1 is needed for a negative feedback loop that engage more RhoA deactivators upon RhoA activation, highlighting the complex regulation of RhoGTPases in space and time.

*For correspondence:
olivier.pertz@unibe.ch

†These authors contributed equally to this work

Competing interest: The authors declare that no competing interests exist.

## Abstract

Emerging evidence suggests that Guanine nucleotide exchange factors (GEFs) and GTPase-activating proteins (GAPs) bind to the cytoskeleton or focal adhesions (FAs), controlling spatio-temporal Rho GTPase activity through feedback mechanisms. We explore such feedback in the Rho-specific GAP Deleted in Liver Cancer 1 (DLC1), which binds to FAs through mechanosensitive interactions. Using a FRET biosensor, we show that DLC1 loss of function leads to globally increased Rho activity and contractility in fibroblasts. Although Rho activity appears macroscopically steady, individual molecules undergo 'signaling flux'—a dynamic cycle of activation and deactivation. To measure this flux, we built a genetic circuit that enables both optogenetic activation of Rho and simultaneous readout of Rho activity. In cells at mechanical steady state, this reveals that DLC1 globally controls the rate of Rho deactivation, both at FAs and at the plasma membrane. Transient induction of local contractility, however, shows DLC1 associating with and dissociating from FAs during their reinforcement and relaxation, which might provide local positive feedback on Rho activity for robust FA disassembly. Together, our results indicate that DLC1 regulates Rho activity both globally at steady state and locally at FAs under tension, highlighting the complexity of spatio-temporal Rho GTPase signaling.

## Introduction

Rho GTPases regulate the cytoskeletal dynamics that power morphogenetic processes at the cell- and tissue level (*Etienne-Manneville and Hall, 2002*). Their activity is tightly controlled in time and space by GEFs and GAPs, which respectively activate and inhibit Rho GTPases. In the GTP-loaded, active state, Rho GTPases bind downstream effectors that regulate cytoskeletal dynamics and other cellular processes. Based on classic experiments that include measuring population-averaged Rho GTPase activity levels, as well as overexpressing dominant positive/negative Rho GTPase mutants, it was proposed that Rac1 controls lamellipodial protrusion, Cdc42 regulates filopodium formation, and RhoA controls myosin-based contractility.

Recent studies in which spatio-temporal Rho GTPase activation dynamics were measured using fluorescence resonance energy transfer (FRET)-based biosensors revealed a much higher signaling

complexity. During cell migration, RhoA, Rac1, and Cdc42 are all activated in membrane protrusions at the leading edge (*Itoh et al., 2002*; *Kraynov et al., 2000*; *Nalbant et al., 2004*; *Pertz et al., 2006*). Further, the three Rho GTPases are activated in specific spatio-temporal sequences during the protrusion/retraction cycles that fluctuate on time and length scales of seconds and single micrometers (*Machacek et al., 2009*). A further level of complexity is that these Rho GTPase activity sequences vary depending on the cell system (*Hu et al., 2022*; *Machacek et al., 2009*; *Martin et al., 2016*). This indicates that complex spatio-temporal Rho GTPase signaling programs regulate processes such as cell migration. Similarly, as for Ras GTPases (*Schmick et al., 2015*), it is now clear that these Rho GTPase signaling patterns most likely result from a precise spatio-temporal regulation of Rho GTPase signaling fluxes controlled by GEFs and GAPs (*Bement and von Dassow, 2014*; *Fritz and Pertz, 2016*; *Miller and Bement, 2009*; *Pertz, 2010*). In this signaling flux, specific GEFs will be locally activated to load Rho GTPases with GTP depending on different cellular inputs. These GTP-loaded GTPases will then subsequently diffuse in the plasma membrane (PM) by virtue of their C-terminal lipid moiety until they encounter a locally activated GAP which eventually deactivates them. The interplay of GEFs, GAPs, and diffusion of Rho GTPases in the PM will ultimately lead to the formation of a Rho GTPase activity pattern. Modeling studies have suggested that sophisticated dynamic signaling behaviors can emerge from the interplay of GEFs, GAPs, and GTPases within a signaling network (*Tsyganov et al., 2012*). Steady-state images of Rho GTPase activity patterns do not provide information about the whole Rho GTPase flux mentioned above.

This signaling complexity is consistent with the large amount of GEFs and GAPs that are ubiquitously expressed in cells (*Fusco et al., 2016*; *Fusco et al., 2016*; *Moon and Zheng, 2003*; *Mosaddeghzadeh and Ahmadian, 2021*; *Müller et al., 2020*; *Rossman et al., 2005*). Recently, a system-wide screen has revealed that many GEFs and GAPS localize to cytoskeletal structures as well as adhesion complexes such as focal adhesions (FAs; *Müller et al., 2020*). This strongly suggests the existence of feedback mechanisms from the cytoskeleton and FAs to Rho GTPase signaling. In this study, it was shown that Rac1-specific GEFs bind to FAs at the front, while Rac1-specific GAPs bind to FAs and at the back of the lamella of fibroblasts. This precise, asymmetric positioning of GEFs and GAPs might then regulate the Rho GTPase flux that produces the wide gradient of Rac activity observed at the leading edge of fibroblasts (*Itoh et al., 2002*; *Kraynov et al., 2000*; *Martin et al., 2016*). Such feedback regulation might allow leading edge Rac1 activity to constantly integrate mechanosensitive inputs from FAs, allowing dynamic regulation of Rac1 activity required to fine-tune cell migration. Another prominent example of such spatio-temporal feedback regulation is the excitable RhoA activity patterns observed in the cortex during cytokinesis in *Xenopus laevis* (*Bement and von Dassow, 2014*), that have also been observed in mammalian cells (*Graessl et al., 2017*). Here, RhoA activity waves that control F-actin wave patterns are spatially regulated by a RhoGAP that is locally regulated by F-actin (*Bement et al., 2015*). Thus, cytoskeletal feedback to Rho GTPase regulation might be crucial to generate dynamic Rho GTPase signaling patterns. Understanding such spatio-temporal feedback regulation is not accessible with classic genetic approaches in which long-term perturbation of Rho GTPases, their regulators, and effectors rapidly result in a new mechanochemical state of the cell that will not be informative about an initial signaling/mechanical state of interest (*Isogai and Danuser, 2018*). Further, it is currently unknown which specific features of spatio-temporal Rho GTPase signaling are regulated to produce signaling patterns. For example, does negative regulation by a GAP solely control Rho GTPase signal termination, or could it also modulate less intuitive parameters such as the rate of activation? Tackling such questions requires new tools to acutely and transiently perturb the Rho GTPase flux to understand its spatio-temporal regulation, directly in single living cells. Note that this experimental paradigm was successful at analyzing the Mitogen-activated protein kinase (MAPK network; *Blum et al., 2019*; *Dessauges et al., 2022*; *Ryu et al., 2015*).

A promising candidate to investigate how mechanical feedback from FAs can spatio-temporally shape Rho GTPase activity is the RhoA-specific GAP Deleted in Liver Cancer 1 (DLC1). DLC1 has been shown to both bind the PM (*Erlmann et al., 2009*), as well as FAs (*Haining et al., 2018*; *Kaushik et al., 2014*). At the PM, interaction with phosphatidylinositol-4,5-bisphosphate (PI(4,5)P2) can regulate GAP activity (*Erlmann et al., 2009*). At the FA, DLC1 interacts with the r7-r8 domains of the FA protein talin in a mechanosensitive fashion (*Gingras et al., 2008*; *Goult et al., 2021*; *Haining et al., 2018*; *Zhao et al., 2022*), as well as with FA kinase (*Li et al., 2011*). When talin is in a state of low mechanical strain, it binds to DLC1 and presumably leads to local RhoA inactivation. However, upon

force application, the r7-r8 domains unfold, leading to dissociation of DLC1 from talin, leading to loss of negative RhoA regulation. This suggests that DLC1 is regulated at FAs by mechanosensitive interactions, which can relay information about the mechanical state of the cell to control RhoA activity.

In this work, we explore the spatio-temporal regulation of RhoA activity by DLC1. We show that knocking out DLC1 in REF52 fibroblasts leads to increased amplitude in RhoA throughout the cell. This is accompanied by an increased contractility that augments FAs and stress fibers (SFs). To explore how DLC1 contributes to Rho GTPase activity fluxes, we built a genetic circuit consisting of an optogenetic actuator to activate Rho with light, and a spectrally compatible biosensor to measure Rho activity. Transient optogenetic recruitment of a Rho GEF domain at FAs or at the PM led to an increase in the rate of activation of Rho in *DLC1* KO versus WT cells at both subcellular localizations. However, we found that in both control and DLC1 null cells, RhoA activation was more efficient at FAs compared to the PM. Further, acute and local manipulation of contractility with the optogenetic actuator revealed that DLC1 dynamically associates/dissociates with FAs under acute mechanical tension/relaxation. Our results suggest a complex mechanism in which DLC1-dependent Rho regulation can occur both at the PM and at FAs, with a dependence on mechanosensitive signals on the latter.

## Results

### DLC1-deficient cells show increased formation of SFs and FAs

To investigate the function of DLC1 in cytoskeletal and adhesion dynamics, we knocked down (KD) DLC1 in rat REF52 fibroblasts via small interfering RNA (siRNA). We also created a knockout (KO) cell line via Crispr/Cas9 that has a frameshift deletion at position 841 of Exon 5 of the gene, resulting in an early in-frame stop codon 100 bp downstream of the cut. Gene editing was validated by sequencing the endogenous locus. We were unable to test expression levels using western blot because of the absence of commercially available antibodies specific for rat DLC1. We then also rescued the *DLC1* KO cells with an mCherry-labeled DLC1 construct using stable transfection with a piggyBac transposase system. REF52 fibroblasts were seeded at subconfluent density onto fibronectin-coated glass bottom wells and allowed to spread for 1 hr before fixation, leading to an isotropic spreading state in control, wild-type (WT) cells. To assess the impact of DLC1 deficiency on the formation of cytoskeletal and adhesion structures, we stained for F-actin (phalloidin) and paxillin (*Figure 1A*) or phospho-myosin light chain (pMLC; *Figure 1B*). Consistent with an expected increase in RhoA activity, we observed a phenotype of increased contractility as documented by increased lamella size, SF content, number of FAs, and increased pMLC. This increased contractility led to the loss of the isotropic spreading observed in WT cells. Rescue with the DLC1 construct reverted the cells to a less contractile phenotype that displayed a lamella of identical size as WT cells, enabling the characteristic isotropic spreading observed in WT cells. Quantification of these images revealed both an increase in lamella size (*Figure 1C*) and FA area (*Figure 1D*) in DLC1 KD and KO cells versus WT and rescue cells (*Figure 1C*). As previously documented in fibroblasts (*Kaushik et al., 2014*), these results indicate that DLC1 feeds into the regulation of contractility, SF, and FA formation in fibroblasts. The less penetrant phenotype observed in KO versus KD suggests that the KO cells might be able to adapt to the long-term absence of DLC1, while KD cells had less than 48 hr to adjust to the new mechanical state. These experiments also validate the *DLC1* KO cells and *DLC1* KO rescued with mCherry-DLC1 that will be used in subsequent experiments.

### DLC1-deficient cells display altered spreading dynamics, lack of polarization, and efficient migration

We investigated cytoskeletal dynamics during spreading using REF52 fibroblasts stably expressing the F-actin marker LifeAct (*Riedl et al., 2008*). Imaging early spreading (e.g. 20 min after plating) revealed no clear differences in edge dynamics in WT versus *DLC1* KO cells (*Figure 2A*). However, kymograph analysis clearly showed that the lamella that characteristically starts directly at the leading edge of the highly contractile REF52 cells (*Martin et al., 2016*; *Martin et al., 2014*) was wider in *DLC1* KO versus WT cells. Later, 1 hr after plating, when the cells are still spreading, lamellipodial protrusion/retraction cycles of wider amplitude were observed in *DLC1* KO versus WT cells (*Figure 2B*, *Figure 2—video 1*), with again a large increase in SF content (*Figure 2B*, right panel). After initial spreading, REF52 fibroblasts break their symmetry and eventually display short episodes of polarized cell migration. Later,

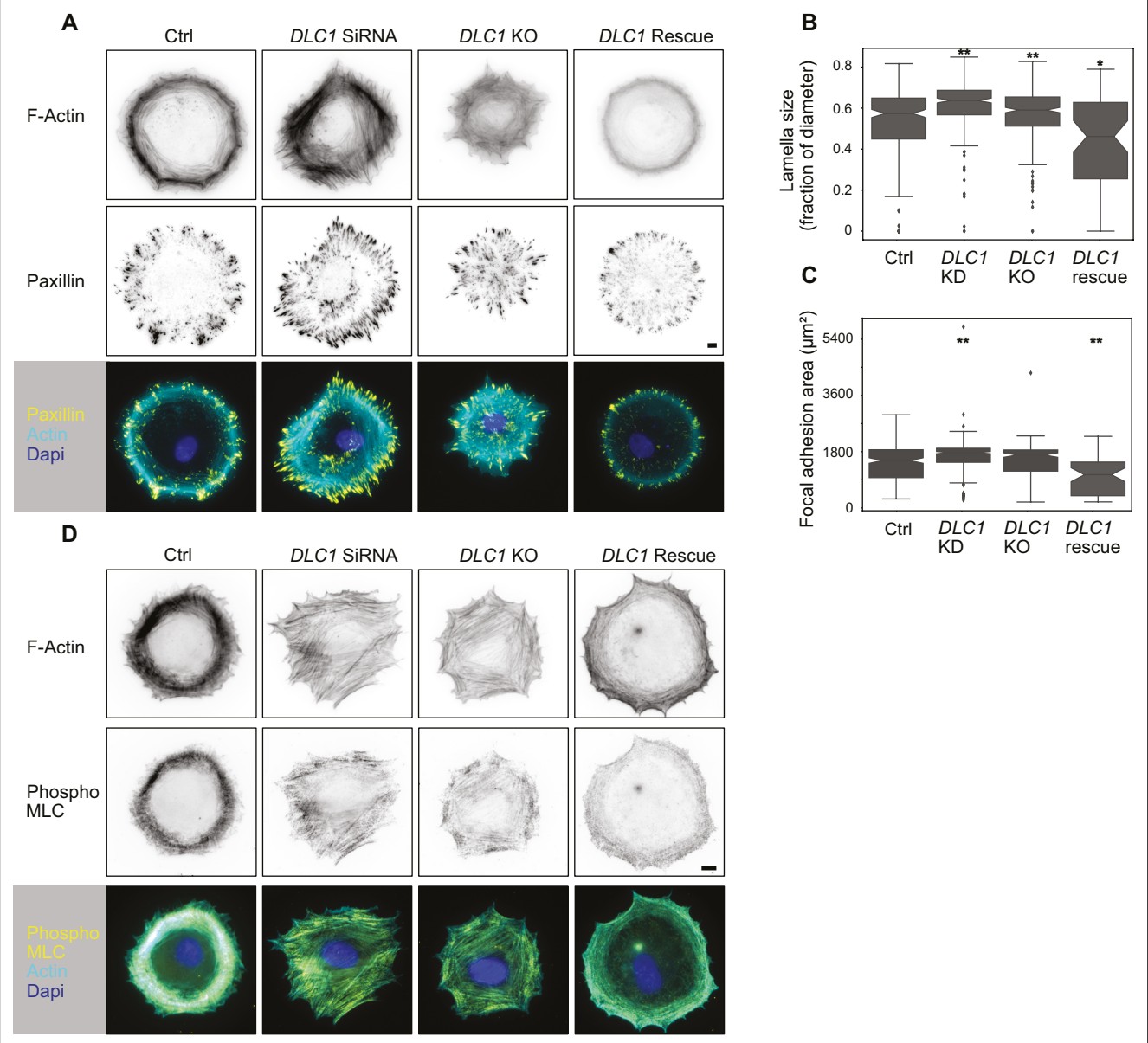

**Figure 1.** Cytoskeletal structures in WT and DLC1 deficient cells. (**A,B**) Representative immunofluorescence images of paxillin and F-actin (phalloidin) (**A**), and pMLC and F-actin (phalloidin) (**B**) immunostains are shown in inverted black and white (ibw) contrast as well as color composites (DAPI signal also included). Scale bar = 10 μm. (**C**) Compared to WT cells (n=243), lamella size is significantly increased in *DLC1* KD cells (n=290, p-adjusted<0.001) and *DLC1* KO cells (n=242, p-adjusted<0.001), while *DLC1* rescue cells (n=57) show significantly lower lamella sizes than both WT (p-adjusted=0.0098) and KO cells (p-adjusted<0.001). (**D**) *DLC1* KD cells (n=290) display a significant increase in the total area of focal adhesions per cell (p-adjusted<0.001) compared to WT cells (n=243), while *DLC1* KO cells (n=242) do not show a significant difference (p-adjusted=0.09). Rescue cells show a significant decrease in the total area of focal adhesions (p-adjusted<0.001). ANOVA plus Tukey's honestly significant difference test.

contractility builds up, leading cells with spindle-like morphology. This process occurs on timescales of multiple hours. To capture these behaviors, we investigated cytoskeletal dynamics over the next 10 hr after reseeding at a slower timescale that precludes observation of lamellipodial dynamics. While WT cells efficiently broke their symmetry and could display short stretches of polarized migration, *DLC1* KO cells displayed robust SFs that hampered symmetry breaking and polarization (*Figure 2C*, *Figure 2—video 2*). *DLC1* KO cells, therefore, rapidly adopted a spindle-like shape. Quantification of the adoption of such a spindle shape phenotype showed that WT cells will assume the spindle-like shape much later than *DLC1* KO cells and often fail to do so entirely within the 10 hr of imaging (*Figure 2D*). Consistent with these impaired cytoskeletal dynamics, *DLC1* KO cells displayed a reduction in both

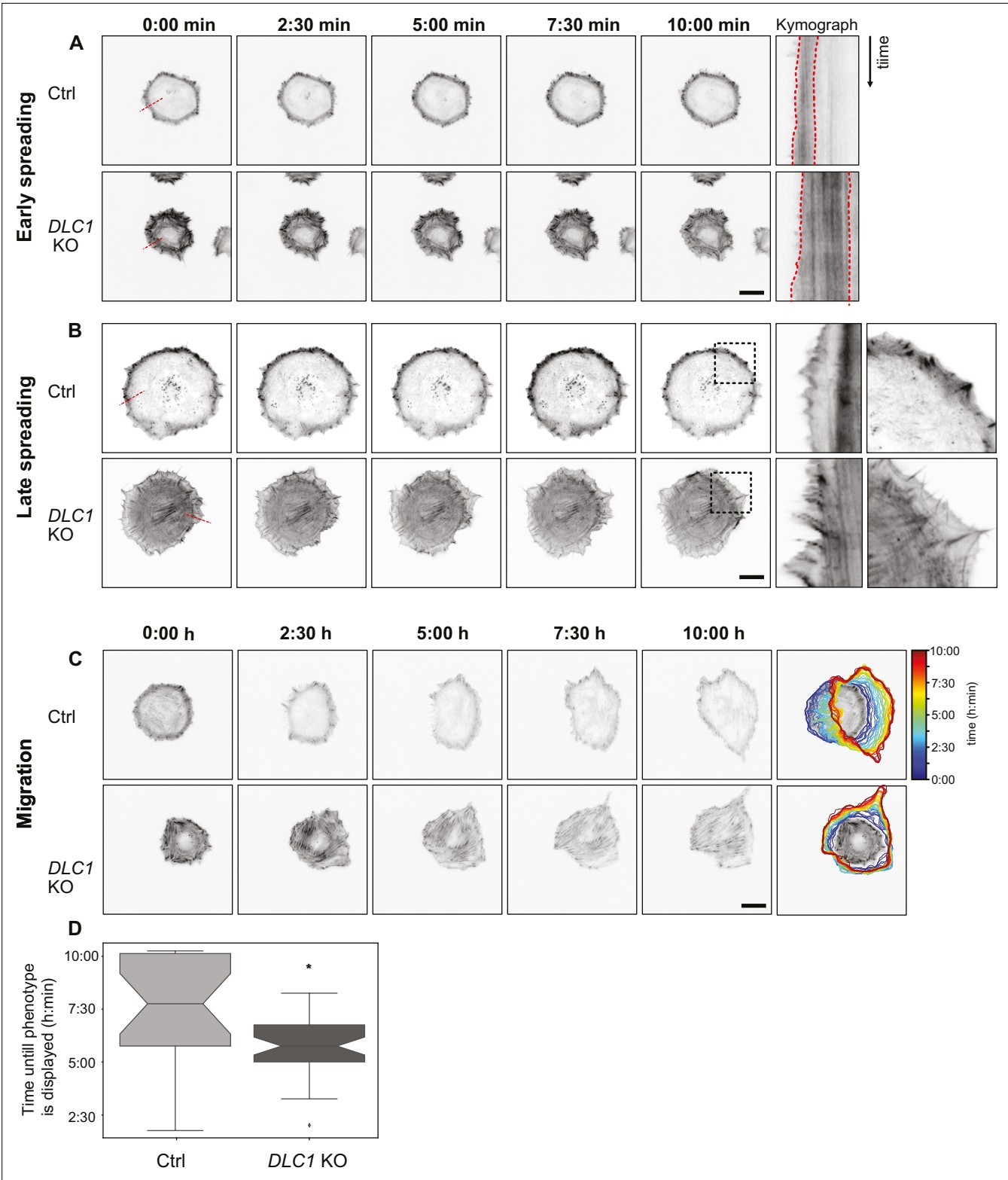

**Figure 2.** F-actin dynamics and cell morphodynamics in WT and *DLC1* KO cells. All time series are representative ibw contrast images of WT and *DLC1* KO cells expressing Lifeact-mCherry. Dotted lines mark the area used for the kymographs shown in the right panels. (**A**) Cells were imaged for 10 min in 10 s intervals during spreading immediately after replating. Dotted red lines display differences in lamella size in WT and *DLC1* KO cells. Time (**B**) One hour after reseeding, when cells are still isotropically spreading, *DLC1* KO cells display increased lamellipodia size and more prominent edge protrusion retraction cycles. High-magnification insets are shown for time point 10 min and show the robust increase of contractility in *DLC1* KO cells. (**C**) Cells were

*Figure 2 continued on next page*

*Figure 2 continued*

imaged for 10 hr with 15 min intervals starting from spreading. WT cells break symmetry and display episodes of polarized motility. KO cells are much less dynamic and reach a contractile phenotype much faster, without being able to polarize. (D) Box plot showing at which time point cells exhibited the elongated phenotype described in (C). *DLC1* KO cells (n=42) exhibited this phenotype a mean of 90 min earlier than WT cells (n=22) p-adjusted=0.002, Student's t-test. 17 additional WT cells failed to develop that phenotype entirely. Scale bars = 20 μm for all images.

The online version of this article includes the following video and figure supplement(s) for figure 2:

**Figure supplement 1.** Cell motility properties in WT versus DLC1 KO cells.

**Figure supplement 2.** Effect of DLC1 overexpression on F-actin cytoskeleton.

**Figure 2—video 1.** F-actin dynamics of WT and DLC1 KO REF52 cells during late spreading (relevant to *Figure 2B*).

https://elifesciences.org/articles/90305/figures#fig2video1

**Figure 2—video 2.** F-actin dynamics and edge dynamics of WT and DLC1 KO REF52 cells during acquisition of a polarized cell migration phenotype over a period of 15 hr (relevant to *Figure 2C*).

https://elifesciences.org/articles/90305/figures#fig2video2

their speed of migration and their ability to migrate directionally in the presence of platelet-derived growth factor (PDGF), a potent stimulator of fibroblast motility (*Martin et al., 2014*; *Figure 2—figure supplement 1*). Further highlighting a role for DLC1 in RhoA-mediated control of contractility, we found that DLC1 overexpression leads to strong reduction in the number of SFs (*Figure 2—figure supplement 2*). Together, these results show that DLC1 loss of function leads to increased contractility at the onset of spreading, which later impedes polarization and directional cell migration.

## DLC1 deficient cells display an increase in global RhoA activity, without any marked differences in RhoA activity pattern

To explore how DLC1 spatio-temporally controls RhoA activity, we used RhoA2G (*Fritz et al., 2013*), a fluorescence resonance energy transfer (FRET)-based biosensor that reports on RhoA activity together with the Lifeact F-actin marker during spreading. As previously observed (*Martin et al., 2014*), a RhoA activity band that correlates with the lamellar contractile actomyosin network of the cell can be observed at the cell edge of WT REF52 fibroblast (*Figure 3A*, *Figure 3—video 1*). In *DLC1* KO cells, we observed global elevation of RhoA activity throughout the cell. Quantification of both the average activity per cell and the local edge RhoA activity revealed increased RhoA activity (*Figure 3B*). Kymograph analysis revealed that the lamellar RhoA activity band remained of constant size during spreading (*Figure 3A*). Examining cells approximately 10 hours after spreading when they had fully spread, and WT cells had adopted a more contractile state, the DLC1 *KO* cells still exhibited a global increase in RhoA activity (*Figure 3C and D*). This increase was, however, less strong than in spreading cells, most likely because of a higher state of contractility in WT well-spread versus spreading cells. We also evaluated the effect of overexpression of an mCherry-tagged DLC1 on RhoA activity (*Figure 3—figure supplement 1*). We observed that mild DLC1 expression, which remains mostly bound to FAs and to the PM and does not compromise cell morphology, leads to a decrease in global RhoA activity. Strong DLC1 expression that leads to a large cytosolic pool of DLC1 further diminishes global RhoA activity but then compromises cell morphology. Together, these results suggest that DLC1 signaling at the PM and at FAs controls global levels of RhoA activity within cells, rather than controlling the spatio-temporal RhoA activity pool observed at the lamella.

## Design of an optogenetic actuator - Rho biosensor to interrogate the Rho GTPase flux

While FRET measurements as shown above provide insight about a steady-state RhoA activity pattern, they cannot probe in detail how a GEF/GAP/GTPase signaling network might affect the Rho GTPase flux which most likely emerges from the regulation of multiple GEFs and GAPs. For example, GAP-mediated negative regulation might control Rho GTPase signal duration, but might also be involved in adjusting the rate of activation. We reasoned that transient, acute optogenetic perturbations of the Rho GTPase flux, followed by measuring Rho activity dynamics, might provide new insights about its regulation. To manipulate Rho activity in single living cells, we re-engineered an iLID-based opto-genetic actuator based on a DH/PH domain of the GEF LARG (*Oakes et al., 2017*; *Figure 4A*). We used a high-affinity nano SspB domain for efficient light-dependent recruitment of the LARG GEF

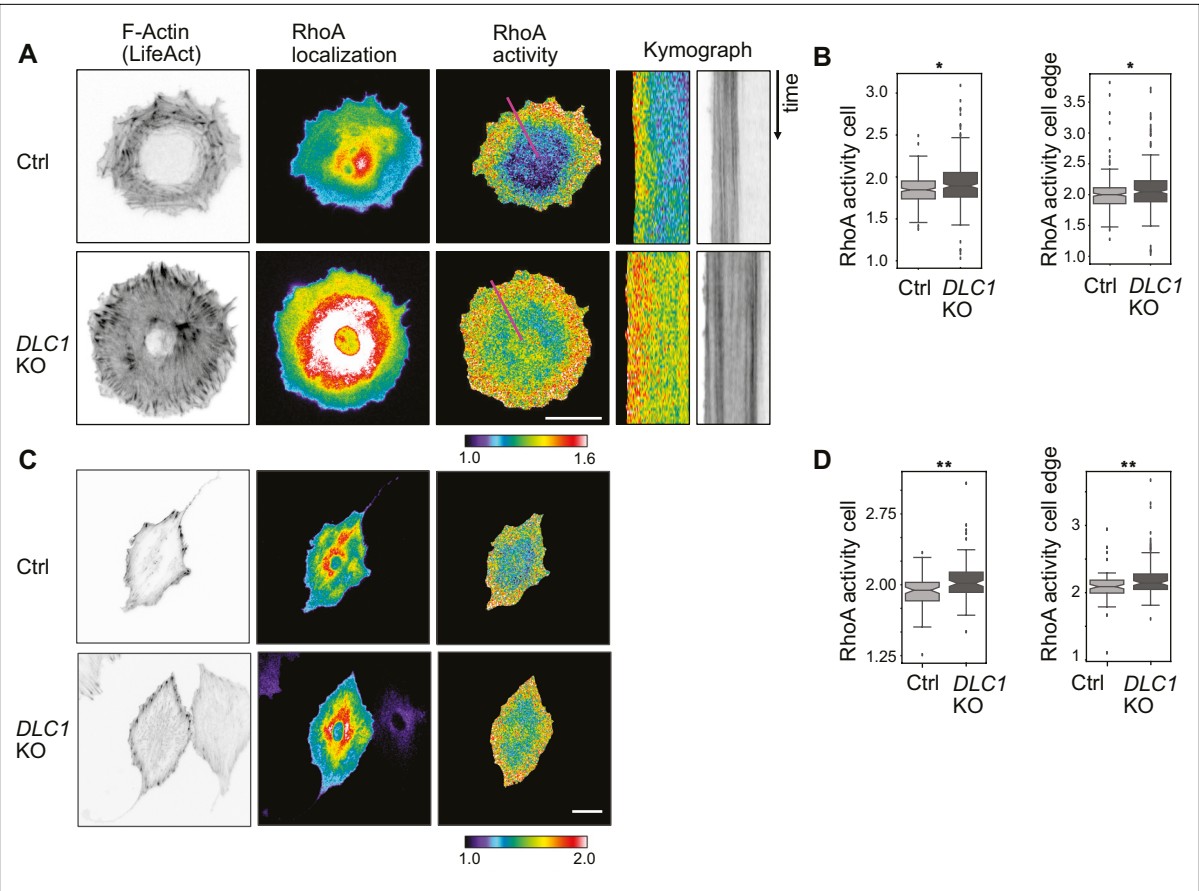

**Figure 3.** RhoA activation dynamics in WT and *DLC1* KO cells. Cells are stably expressing Lifeact-mCherry and the RhoA2G FRET sensor. RhoA localization images show RhoA2G localization that is identical to RhoA. RhoA activity images display the computed FRET ratio. Images are color-coded according to the normalized scales shown below the panels. (**A**) Representative images of WT and *DLC1* KO cells during spreading. Kymographs for the violet lines are shown in the right panels. Note the RhoA activity band maintains constant width during spreading at the periphery in WT cells. Note increased global RhoA activity in *DLC1* KO cells, with maintenance of a similar RhoA band pattern at the cell edge. (**B**) Box plots of FRET ratio averaged over the whole cell (right panel) or a ROI placed at the cell edge (left panel). This shows that during spreading, *DLC1* KO cells (n=149) have an increased total RhoA activity compared to WT cells (n=114, p=0.005). In addition, the FRET ratio at just the cell edge is increased as well (p=0.011). (**C**) Representative images of WT cells and DLC1 KO cells that have transitioned in a contractile state 12 hr after plating. No difference in RhoA activity pattern can be observed between WT and *DLC1* KO cells, although the latter still display slightly higher global RhoA activity levels. (**D**) Box plots of FRET ratio averaged over the whole cell (right panel) or a ROI placed at the cell edge (left panel). This shows that contractile *DLC1* KO cells (n=103) have an increased total RhoA activity (p<0.001) and edge FRET ratio (p<0.001) compared to WT cells (n=82) compared to WT cells (n=82). Students t-test. (**A,B**) Scale bars = 20 µm.

The online version of this article includes the following video and figure supplement(s) for figure 3:

**Figure supplement 1.** Effect of DLC1 overexpression on RhoA activity.

**Figure 3—video 1.** F-actin and RhoA activity dynamics in WT and DLC1 KO REF52 cells during late spreading (relevant to *Figure 3A*).
https://elifesciences.org/articles/90305/figures#fig3video1

domain to the iLID anchor. To better focus optogenetic activation, we also fused the iLID anchor to a stargazin membrane anchor that slows down its diffusion (*Natwick and Collins, 2021*). Both the stargazin-iLID anchor and the SspB-LARG domains were separated by a P2A self-cleaving peptide, allowing for equimolar expression of both units from a single operon. We refer to this construct as optoLARG. To interrogate the Rho GTPase flux, we engineered a REF52 line that stably expressed the optogenetic construct as well as a rhotekin-based G protein binding domain (rGBD) that reports on Rho activation (*Mahlandt et al., 2021*). The rGBD probe is labeled with a tandem dimeric Tomato (tdTomato) fluorophore that is spectrally orthogonal to optogenetic activation. We used a truncated CMV promoter to warrant low expression level of the rGBD probe to avoid dominant negative effects of the construct through inhibition of Rho signaling. We also expressed a spectrally orthogonal far-red

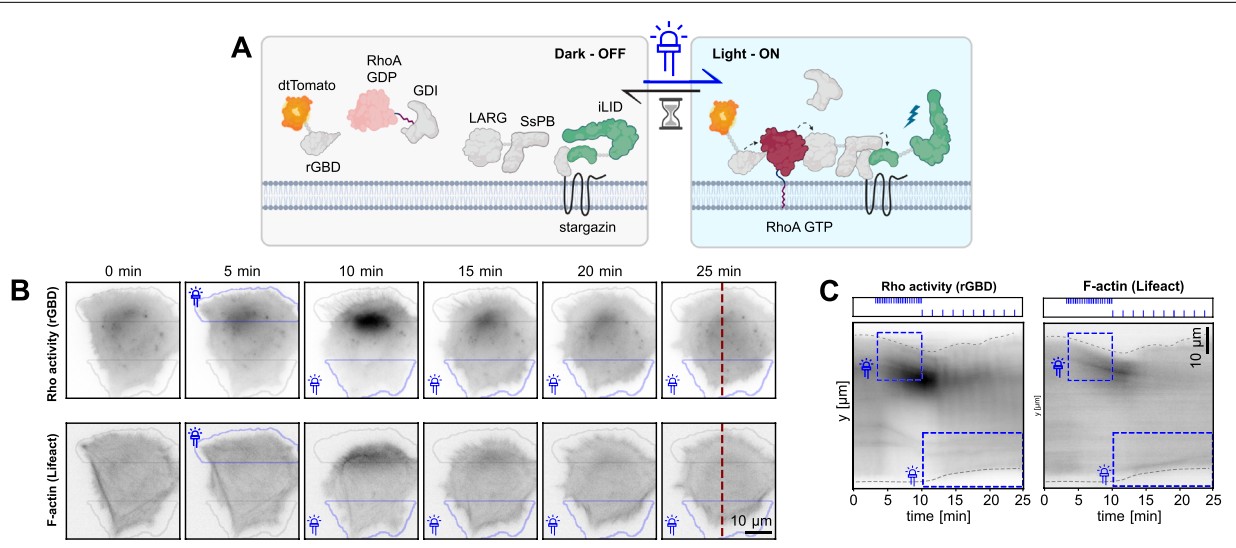

**Figure 4.** An optogenetic actuator - Rho biosensor circuit to probe Rho GTPase flux. (**A**) Schematics of the optogenetic actuator - Rho biosensor to measure Rho GTPase flux. OptoLarg is based on an iLID system which does not interact with Ssb-LARG in the dark state. The iLID module is anchored to the PM by a Stargazin anchor that displays slow diffusion, allowing better focusing of optogenetic activation. Upon light exposure, ssb-LARG is locally recruited to the plasma membrane, activating Rho. Rho activity is measured by a rGBD effector binding domain. (**B**) Time series of REF52 cells locally stimulated with light first in a ROI at the cell top with a high stimulation frequency, and then with a ROI at the cell bottom with a low stimulation frequency. Light pulses have the same intensity. Thick and thin blue thunder symbols represent high and low optogenetic stimulation. A red dotted line is used for the kymograph shown in (**C**). (**C**) Kymographs of cells in (**B**). Light pulse stimulation regimes of top and bottom ROIs are shown in the upper box. Blue dotted boxes indicate the region and length of the stimulation. Note how intense optogenetic stimulation in the top ROI leads to Rho activity, assembly of contractile F-actin structures, and robust edge retraction. Upon removal of the light input, the Rho activity and F-actin resume, and edge protrusion occurs again. Lower optogenetic stimulation in the bottom ROI leads to much lower Rho activity and F-actin structures, as well as lower edge retraction. (**B,C**) Scale bars = 10 µm.

Lifeact-miRFP construct that reports on F-actin dynamics. Because rGBD cannot distinguish between the activity of the RhoA, RhoB, and RhoC isoforms, we refer to any rGBD measurement as Rho activity. This is in marked contrast with RhoA2G, the FRET biosensor we used above that is specific for RhoA (*Fritz et al., 2013*), but is not spectrally compatible with optoLARG. Note that both biosensors are able to measure active Rho GTPase pools during robust morphogenetic events such as tail retraction (*Fritz et al., 2013*; *Mahlandt et al., 2021*), while rGBD misses leading edge Rho GTPase activity pools widely observed using FRET probes (*Machacek et al., 2009*; *Martin et al., 2016*; *Pertz et al., 2006*).

We then used a digital micromirror device (DMD) to spatially shine repetitive pulses of blue light delivered at different frequencies on two distinct regions of interest (ROI) at the edge of fibroblasts. When light pulses were delivered at high frequency, we observed robust Rho activation at the edge, which correlated with assembly of contractile F-actin structures and potent edge retraction (*Figure 4B*, kymograph shown in 4C, *Figure 5—video 1*). Removal of the blue light pulses led to a subsequent decrease in Rho activity and reversion of the edge to a protrusive state. Further, stimulating an ROI at the opposite edge of the cell with a lower light pulse frequency led to a lower Rho activation and less robust edge retraction. These results indicate that depending on the strength of the light input, opto-LARG can locally, reversibly, and quantitatively control Rho activity in single living cells on timescales of seconds, allowing us to acutely manipulate the Rho GTPase flux.

## Optogenetic interrogation of the Rho GTPase flux in WT and *DLC1* KO cells

Because DLC1 might both regulate the Rho GTPase flux both at the PM (*Erlmann et al., 2009*) and at FAs (*Haining et al., 2018*), we performed a series of experiments using optoLARG to transiently activate RhoA with the expectation of observing different perturbations in Rho activity fluxes in control versus *DLC1* KO cells. For that purpose, we produced WT and *DLC1* KO REF52 lines expressing optoLARG, the tdTomato-rGBD biosensor and miRFP-paxillin as a readout for FAs. We first sought to locally activate RhoA by illuminating large ROIs at edges and evaluate any difference in Rho activity

patterns using the spectrally compatible rGBD biosensor in WT versus *DLC1* KO cells. We, however, were not able to identify a threshold optogenetic light input that would induce different RhoA activity patterns and edge dynamics in WT versus the more contractile *DLC1* KO cells. We observed that the same light input could induce edge retraction in both WT and *DLC1* KO cells in a non-predictable manner. This might be due to the spatial heterogeneity of the mechanical states within cells, as well as small differences in optoLARG and rGBD expression levels in the stable cell lines. To address this caveat, we reasoned that we should activate Rho using optoLARG with a more subtle modality by illuminating only small ROIs and measuring small local fluctuations of Rho activity using rGBD without inducing large changes in mechanical states. Further, to dissect potential regulation of *DLC1* GAP activity by PI(4,5)P2 at the PM (*Erlmann et al., 2009*) versus mechanosensitive interactions at the FA (*Haining et al., 2018*), we decided to make measurements on FAs but use ROIs outside of

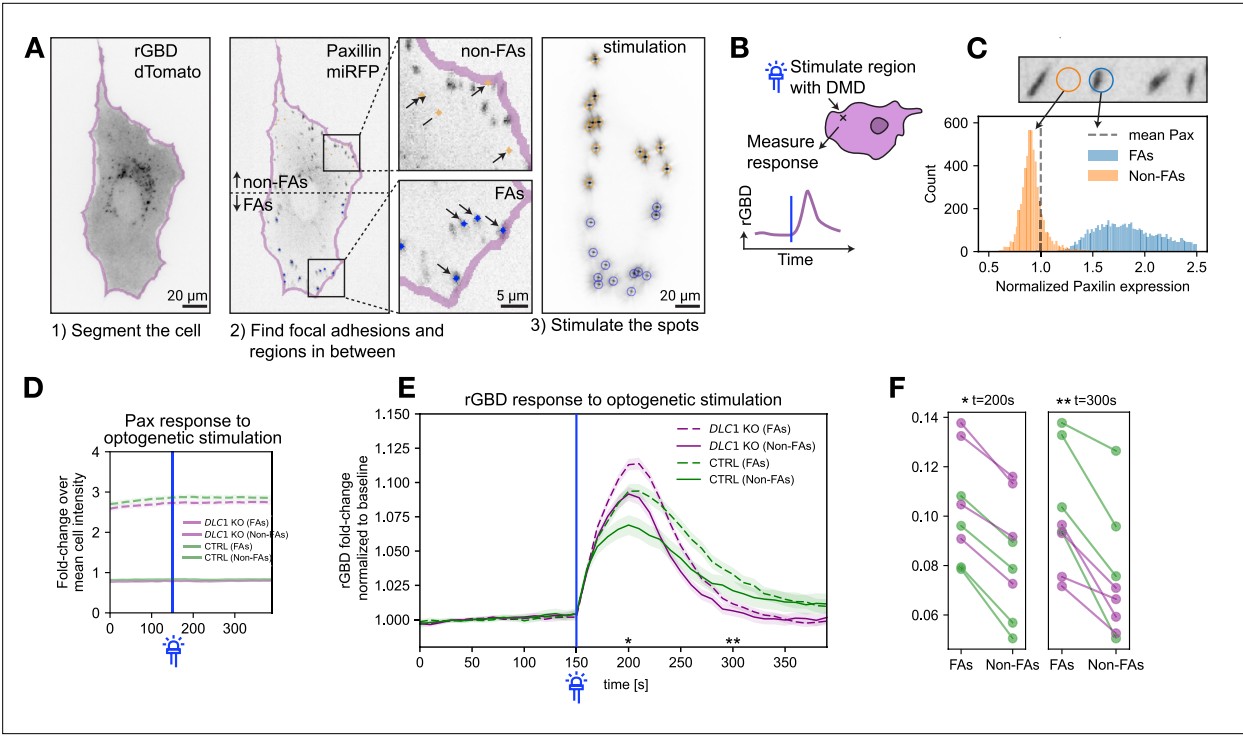

**Figure 5.** Rho GTPase activation kinetics in WT versus DLC1 KO cells. (**A**) Computer vision pipeline and experiment: (1) a cell is segmented using the rGBD-dTomato channel. (2) FAs and non-FAs are detected using the paxillin channel (see Materials and methods). ROIs on FAs (blue dots, shown in the bottom left panel) and ROIs on non-FA regions in between (orange dots, shown in the upper left panel) are selected for stimulation. (3) Image of the stimulation pattern (green channel) shows that the calibrated DMD can stimulate the regions with high spatial precision (image is overexposed to show the diffraction pattern of the mirrors). (**B**) The DMD is used to stimulate ROIs (FAs or non-FAs) with a pulse of blue light (blue line). rGBD signal fluctuations are then measured in the ROIs. (**C**) Distribution of paxillin-miRFP intensities in the FA and non-FA ROIs normalized to the mean paxillin-miRFP intensity of the whole cell shows that our segmentation pipeline accurately identifies FA and non-FA ROIs. (**D**) Paxillin-miRFP fluorescence of stimulated ROIs, normalized to the mean paxillin intensity of the whole cell. Median and 99% CI are shown. The intensity of paxillin does not change upon stimulation, indicating that the stimulation is too weak to trigger reinforcement of focal adhesions (compare to *Figure 7A–D*). (**E**) Normalized and averaged rGBD fluorescence fluctuations upon ROI optogenetic stimulation. For each stimulated ROI, the fold change to the baseline (average activity from 0 to 150 s before optogenetic stimulation) is calculated. Median and 99% CI are shown. Regions on top of focal adhesions have a larger fold change in rGBD activity than regions between focal adhesions. *DLC1* KO cells have a larger rGBD fold change in the initial time after stimulation (150–200 s), but then also fall back down to the baseline quicker. (KO FA: N=3643, KO non-FA: N=3643, WT FA: N=2144, WT non-FA: N=2,144. Mean number of regions per cell: ~16.90, Cells WT = 321, *DLC1* KO = 431). The same cell can appear multiple times in the experiment, but with a relaxation time in between and new stimulation regions. (**F**) The different dynamics described in (**E**) can be robustly observed in technical replicates of the experiment. In all replicates, the rGBD recruitment is higher in FAs vs non-FAs, and we see the trend of faster accumulation and faster return to baseline of rGBD in *DLC1* KO vs WT cells.

The online version of this article includes the following video and figure supplement(s) for figure 5:

**Figure supplement 1.** Quantification of optoLARG recruitment dynamics to subcellular locations.

**Figure 5—video 1.** Characterization of the optoLARG/rGBD genetic circuit (relative to *Figure 5*).

https://elifesciences.org/articles/90305/figures#fig5video1

FAs (non-FAs ROIs) to probe the PM. For that purpose, we built an image analysis pipeline to first segment a cell in a field of view and to then segment FAs based on their fluorescent intensity, size, and localization at the cell periphery using the paxillin-miRFP signal (*Figure 5A*). We then used these FA segmentations to illuminate small ROIs within FAs or in neighboring non-FA ROIs with one pulse of blue light using the DMD and evaluated rGBD fluorescence fluctuations at this location for a couple of minutes until Rho activity returned to baseline (*Figure 5B*). This optogenetic stimulation had no visible effect on edge retraction or FA morphodynamics, as observed when stimulating larger ROIs. Quality control of our automated segmentation revealed high versus low paxillin signals in FA versus non-FA ROIs (*Figure 5C*). This experiment was performed on multiple cells, allowing us to average out any experimental noise due to small differences in expression levels of optoLARG or rGBD. These data were then averaged for FA and non-FA ROIs. We observed that this pulsed optoLARG light input led to a local transient of Rho activity for approximately 2.5 min in WT cells when FAs or the PM were stimulated (*Figure 5D*). To quantify observed activation ($k_{ON}^{obs.}$) and deactivation ($k_{OFF}^{obs.}$) rates, we defined a 10–90% time window of the activation and deactivation phases and computed the mean derivative over each window. The identical optoLARG input led to an increased observed rate of Rho activation leading to higher amplitude when an FA versus a non-FA ROI was stimulated ($k_{ON}^{obs.}$: 0.0132 (non-FA) <0.0176 (FA)). In *DLC1* KO cells, we observed increased Rho activation rate that also led to augmented Rho activity amplitude in comparison with WT cells in both FA and non-FA ROIs ($k_{ON}^{obs.}$: 0.0195 (non-FA) <0.0221 (FA)). This was also accompanied by faster deactivation kinetics of RhoA ($k_{OFF}^{obs.}$: 0.0060 (non-FA KO)<0.0068 (FA KO), $k_{OFF}^{obs.}$: 0.0042 (non-FA WT)<0.0054 (FA WT)). These data suggest that under weak transient GEF input, DLC1 regulates the rate of Rho activation and peak amplitude rather than signal duration. Note that $k_{ON}^{obs.}$ and $k_{OFF}^{obs.}$ here refer to the rates observed in the biosensor signal, which reflect the combined effects of underlying activation and deactivation processes rather than parameters of GEF/GAP activity. This can occur both at the PM and at FAs. The finding that *DLC1* KO also leads to faster deactivation suggests the existence of compensatory negative feedback that points to complex modalities of spatio-temporal regulation that most likely involves additional GEFs/GAPs. In both WT and *DLC1* KO cell lines, RhoA activation rates and peak levels were higher at FAs than in non-FA regions. To exclude that this difference could be an artifact from differential recruitment of the optogenetic tool, we directly measured optoLARG localization using a cell line expressing mScarlet3-optoLARG and miRFP-paxillin, which allowed imaging of optoLARG recruitment without activation. We repeated the experiment from *Figure 5*, this time quantifying optoLARG intensity instead of RhoA activity and found no significant difference in recruitment kinetics between FAs and non-FAs (*Figure 5—figure supplement 1A and B*), excluding tool recruitment as a confounding factor. These results instead suggest that FAs provide a subcellular environment that favors GEF-mediated activation of Rho activity compared to non-FA regions. The recruitment and dissociation dynamics of optoLARG were well described by an inverse exponential function during association and an exponential decay function during dissociation (*Figure 5—figure supplement 1C*). The measured deactivation half-life was 10.8 s, which is faster than previously reported values in 2016 (t½ OFF = 18 ± 2 s in vitro; 32.0±3.7 s for membrane localization in cells; *Hallett et al., 2016*). Our experimental design differs from previous reports in that activation was restricted to small regions of the cell, whereas prior studies used whole-cell illumination. The accelerated decay we observed may therefore result from additional diffusion of optoLARG out of the locally stimulated regions.

The experimental data revealed distinct activation and deactivation kinetics between WT and *DLC1* KO cells, suggesting altered regulatory feedback within the RhoA signaling network. We next sought to investigate the mechanistic basis of these differences by mathematical modeling.

## Modeling suggests loss of negative RhoA autoregulation in *DLC1* KO cells

To quantitatively describe the regulatory dynamics of RhoA signaling following transient, localized optogenetic stimulation at FAs and the PM in WT and *DLC1* KO cell lines, we constructed a system of ordinary differential equations (scheme in *Figure 6A*). This modeling framework allows us to disentangle the underlying activation and deactivation rates from the apparent rates observed with the biosensor and infer qualitative regulatory differences between WT and KO cells. The experimental measurements of optoLARG recruitment were first used to fit a simple linear model of protein activation and deactivation, enabling estimation of the kinetic parameters governing optoLARG dynamics.

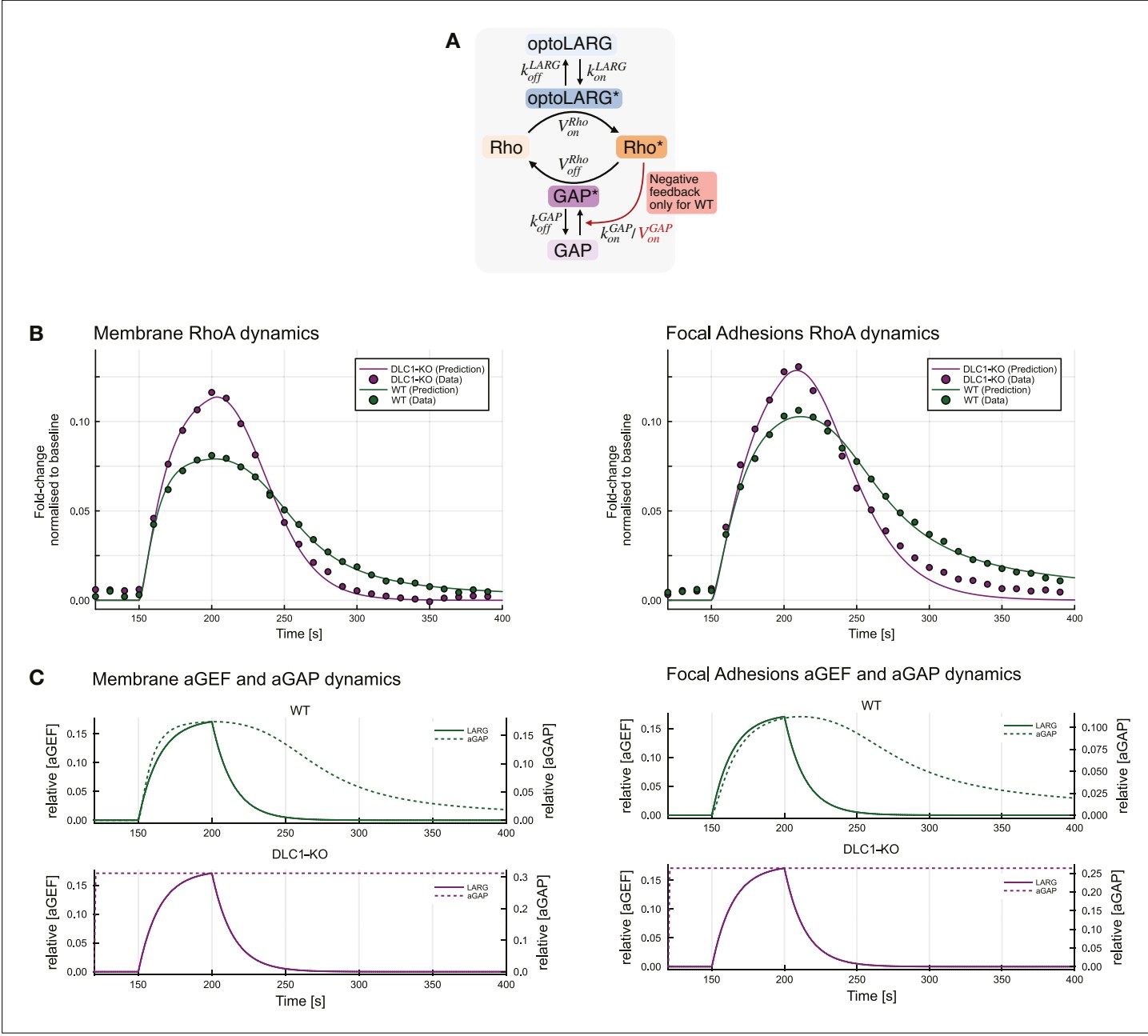

**Figure 6.** Mathematical Model of RhoA differential activation/deactivation dynamics in WT vs. *DLC1* KO cells. (**A**) Schematics of RhoA activation/deactivation system that correspond to the ODE model of RhoA dynamics. The light pulse increases GEF activation rate, which in turn increases RhoA activation. Active RhoA activates GAP, negatively auto-regulating itself. RhoA deactivation is mediated by active GAPs. A detailed description of the modeling approach is provided in the Methods section. (**B**) Results after fitting the ODE model to the data for both RhoA dynamics at the membrane (left panel) and at focal adhesions (right panel). WT and *DLC1* KO conditions were fitted separately in each case. The fold change to the baseline of averaged and normalized rGBD intensity was taken as a datapoint for each acquisition frame. (**C**) Illustration of the dynamics of the other model species that are active GEF and active GAP after the light pulse trigger. The left panel shows the dynamics at the membrane, and the right panel shows the dynamics at focal adhesions.

The online version of this article includes the following video for figure 6:

**Figure 6—video 1.** DLC1 dynamics at FAs during an optoLARG-induced contractility pulse (relevant to *Figure 6A–C*, cell denoted by pink arrow).
https://elifesciences.org/articles/90305/figures#fig6video1

Based on this fit, the active GEF profile (GEF*) was treated as a fixed, time-dependent input to the downstream model of RhoA activity. The remaining regulatory interactions, including RhoA activation by GEFs, RhoA inactivation by GAPs, and RhoA-dependent GAP activation, are captured in the following system using loosely constrained parameters:

$$\frac{d\text{LARG}^*}{dt} \approx k_{\text{on}}^{\text{LARG}} \cdot f_{\text{signal}}(t) - k_{\text{off}}^{\text{LARG}} \cdot \text{LARG}^*$$

$$\frac{d\text{GAP}^*}{dt} \approx \frac{V_{\text{on}}^{\text{GAP}} \cdot \text{Rho}^*}{Km^{\text{GAP}} + \text{Rho}^*} - k_{\text{off}}^{\text{GAP}} \cdot \text{GAP}^*$$

$$\frac{d\text{Rho}^*}{dt} \approx \frac{V_{\text{on}}^{\text{Rho}} \cdot \text{LARG}^*}{Km_{\text{on}}^{\text{Rho}} + \text{LARG}^*} - \frac{V_{\text{off}}^{\text{Rho}} \cdot \text{GAP}^* \cdot \text{Rho}^*}{Km_{\text{on}}^{\text{Rho}} + \text{Rho}^*}$$

$$k_{\text{on}}^{\text{LARG}} = 0.0123462$$

$$k_{\text{off}}^{\text{LARG}} = 0.0700499$$

Michaelis-Menten kinetics were used to model RhoA activation by active optoLARG (LARG*), inactivation by active GAPs (GAP*), and a RhoA-dependent feedback on GAP activation (Rho*) (*de Seze et al., 2023*). GAP inactivation was approximated as a first-order process. Given the transient and low-amplitude nature of the optogenetic stimulus, we assumed that the inactive forms of optoLARG, GAP, and Rho remained in large excess throughout the simulation. This allowed us to simplify the model by treating the concentrations of the inactive species as effectively constant over time. The resulting system of ODEs was solved and fitted to the experimental data using gradient-based optimization in Julia (*Figure 6B and C*). We additionally assessed the structural identifiability of all parameters. Full implementation details and the results of the structural identifiability analysis are provided in the Methods section.

In WT cells, the model provides a strong fit to the experimental data and supports the use of canonical Michaelis-Menten kinetics for RhoA activation and inactivation, as well as the negative feedback on GAP. The results indicate that RhoA activity is dynamically regulated by active GAPs, which are themselves activated by RhoA, forming a negative feedback loop. This feedback ensures that RhoA activation remains transient and self-limiting. The fitted parameters reflect saturable enzyme-substrate interactions, consistent with standard biochemical expectations.

In contrast, the behavior observed in *DLC1* KO cells reveals a disruption of this feedback mechanism. When fitted with the same model structure, the parameter for RhoA-dependent GAP activation converges to zero, indicating loss of negative feedback. Instead, the system appears to rely on a constitutively active or pre-localized GAP-like component that inactivates RhoA at a fixed rate. This configuration eliminates feedback modulation and produces RhoA dynamics that are governed by constitutive GAP activity, possibly due to the global GAP activity of multiple RhoA-specific GAPs in the cell.

A limitation we encountered in our modeling is that the fitted parameter magnitudes happened to depend on the bounds imposed during optimization. Structural identifiability analysis confirmed that all parameters are theoretically globally identifiable given the model structure and observables. Yet in practice, some fitted values were found to vary with the parameter bounds chosen. While some parameters (e.g., responsible for RhoA activation or RhoA-dependent GAP activation) were robust across fits, others varied without significantly affecting model performance. This reflects the distinction between structural and practical identifiability (*Gutenkunst et al., 2007*; *Lam et al., 2022*) where the robustness and uniqueness of fitted parameters can be limited by the experimental data. We therefore limited our attention to qualitative differences in regulation (feedback present vs. absent), rather than absolute numerical parameter values.

## DLC1 differently interacts with FAs depending on their mechanical states

Our finding that we did not observe striking patterns of RhoA activity in the vicinity of FAs in spread cells challenges the idea that mechanical inputs regulate Rho activity in this specific regime of mechanical forces. We speculated that the DLC1/talin system in FAs is only mechanosensitive to stronger mechanical inputs. To evaluate FA-DLC1 interactions, we imaged fibroblasts rescued with mCherry-DLC1 and miRFP-paxillin as a marker for FAs.

To characterize force-dependent interactions of FAs with DLC1 in regimes of strong mechanical perturbations, we engineered our mCherry-*DLC1* rescue cells to stably express the optoLARG system, as well as the miRFP-paxillin construct. We then used the DMD to activate Rho-mediated contractility locally and transiently in small ROIs containing FAs with a standardized light input. The transient light pulse induced either of the two FA behaviors: light-mediated FA reinforcement, as evidenced by a local increase in paxillin density, was followed by 1. FA relaxation (and decrease in paxillin density) upon light removal (*Figure 7A–C*, quantified in *Figure 7D*, *Figure 6—video 1*), or 2. FA rupture (*Figure 7—figure supplement 1A–C*). This was accompanied by a striking simultaneous decrease in DLC1 intensity at the FA during FA reinforcement, and a reassociation of DLC1 during FA relaxation upon light input removal. If the light input led to FA rupture, then the DLC1 signal gradually decreased along FA reinforcement until the FA ultimately ruptured in an all-or-nothing manner. FAs in the same cell that were not subjected to the light input displayed normal dynamics (*Figure 7—figure supplement 1D–F*), indistinguishable from cells without the optoLARG construct: DLC1 binds to FAs at the onset of assembly, with local DLC1 density increasing alongside paxillin density and diminishing concomitantly during disassembly until the FA is fully resolved (*Figure 7—figure supplement 2A–F*). These results indicate that DLC1 dynamically dissociates and associates during acute FA reinforcement and relaxation specifically in response to a strong mechanical input.

## Discussion

A key question in Rho GTPase biology is how multiple GEFs and GAPs control spatio-temporal Rho GTPase signaling patterns to regulate cytoskeletal dynamics that power morphogenetic processes such as cell migration. The recent finding that many GEFs and GAPs themselves bind cytoskeletal and adhesion structures (*Müller et al., 2020*) strongly suggests direct feedback from the cytoskeleton to Rho GTPase signaling. Self-organizational properties of cytoskeletal and adhesions structures might therefore contribute to spatio-temporal control of Rho GTPase signaling. Because DLC1 has been proposed to bind to FAs in a mechanosensitive manner (*Haining et al., 2018*), we decided to study it as a prototypical example of an adhesion feedback to Rho GTPase signaling. Using novel technologies that allow for acute spatio-temporal perturbations, we provide some new insights about how Rho GTPases are spatio-temporally controlled by a RhoGAP that integrates mechanosensitive inputs.

We found that in spreading fibroblasts, *DLC1* KO globally augments contractility, as illustrated by a robust increase in SFs and FAs (*Figure 1*). Evaluation of F-actin dynamics immediately after spreading, when global contractility levels are still low, reveals that *DLC1* KO fibroblasts rapidly assemble a wider contractile lamella than WT cells, suggesting aberrant increase of myosin-based contractility already at an early stage of spreading (*Figure 2A*). One hour after cell plating, when cells are well spread but still retain an isotropic shape, we observed a global increase in contractility in *DLC1* KO versus WT cells (*Figure 3B*). On a timescale of multiple hours after spreading, when WT cells break symmetry, display polarized cell migration episodes, and finally adopt a contractile state leading to a spindle morphology, we find that *DLC1* KO cells transition directly to the contractile phenotype without being able to polarize (*Figure 2C and D*). Thus, DLC1 loss of function leads to a global increase of aberrant contractility.

We had previously shown that REF52 fibroblasts display a wide band of RhoA activity that correlates with the lamellar contractile myosin network during spreading (*Martin et al., 2016*; *Martin et al., 2014*). We found that *DLC1* KO leads to a global increase in RhoA activity throughout the cell, without, however, markedly modifying the spatial band of RhoA activity at the lamellum (*Figure 3A and B*). Similar results were observed when cells were allowed to further polarize and assemble a robust actomyosin cytoskeleton (*Figure 3C and D*). Further, we observed that mild DLC1 expression, which can still mostly be titrated by FAs and the PM without spilling out into the cytosol, is already able to strongly downregulate global RhoA activity levels (*Figure 3—figure supplement 1*). These findings suggest that in spread cells, DLC1 mainly sets the overall baseline of RhoA activity, while the spatial distribution of RhoA activation, such as observed at the lamellar band observed in spreading cells, appears to be maintained by other, spatially restricted GEFs/GAPs that control localized RhoA pools at protrusive edges and filopodia (*Fritz and Pertz, 2016*; *Machacek et al., 2009*; *Pertz et al., 2006*), within the lamella (*Martin et al., 2016*; *Martin et al., 2014*). Similarly, additional GEFs/GAPs might be specifically spatio-temporally regulating the cytokinetic furrow (*Basant and Glotzer, 2018*; *Bement et al., 2015*). Note that the classic genetic perturbation paradigm is only of limited value to

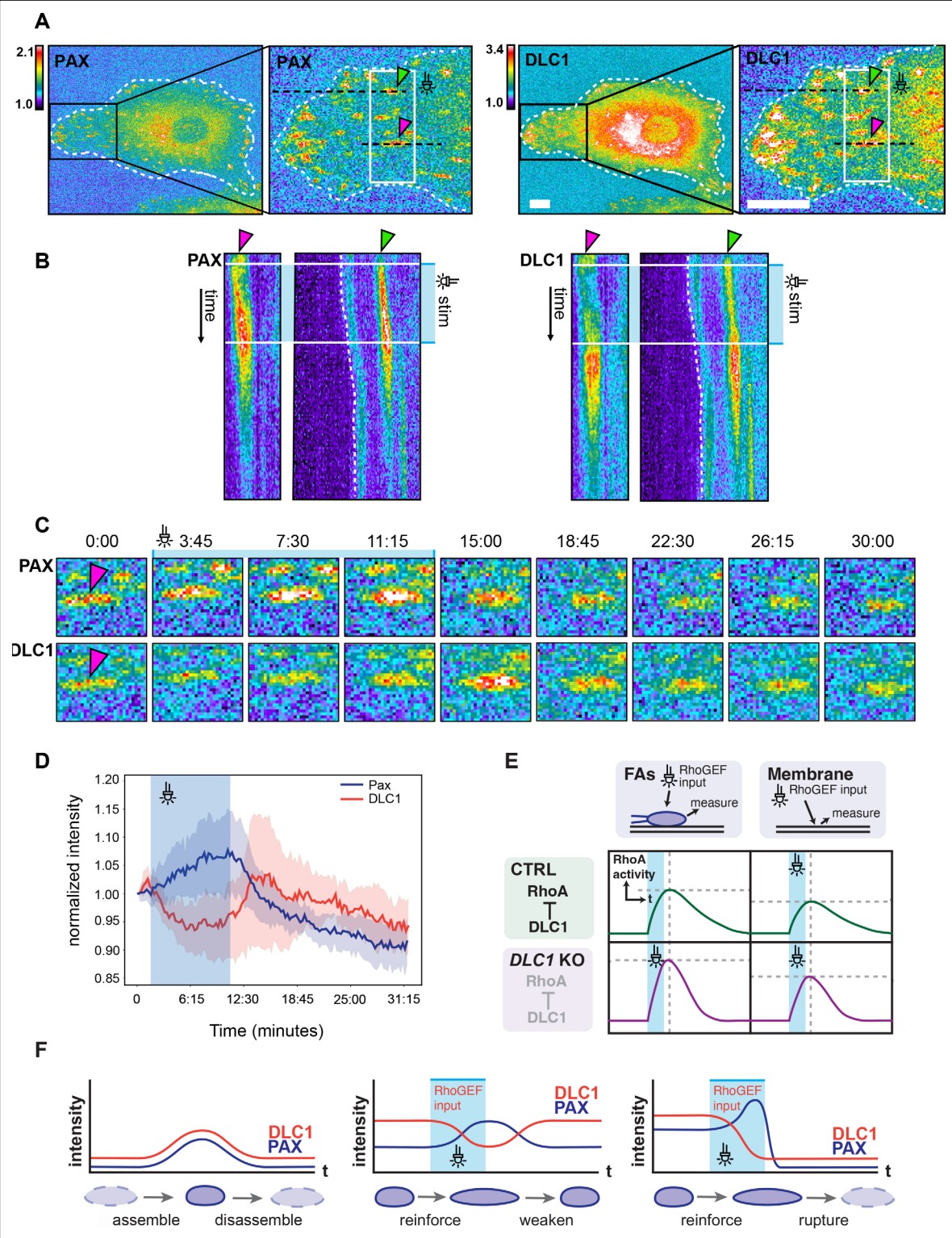

**Figure 7.** Optogenetic control of force-dependent DLC1 interactions with Fas. (**A**) Color-coded fluorescence micrographs of REF52 fibroblasts expressing miRFP-paxillin (left) and mCherry-DLC1 (right) and the optoLARG construct (not shown). The black boxes indicate the area used for close-up images in (**B**). The white boxes indicate the ROI for optogenetic illumination. Selected FAs denoted by the pink and green arrowheads. The black dotted lines were used for the kymograph in **B**. (**B**) Kymographs showing two selected FAs, the gray box indicates the time at which optogenetic stimulation has been applied. Optogenetic stimulation is applied on select ROIs placed over FAs with 50 ms light pulses per frame (every 15 s) for a duration of 7.5 min. (**C**) Close-up time series of paxillin and DLC1 signals at a single FA, denoted by the respective arrowheads. Scale bars = 10 μm.

*Figure 7 continued on next page*

*Figure 7 continued*

(**D**) Quantification of mCherry-DLC1 and miRFP-paxillin fluorescence signals during optoLARG-mediated control of FA reinforcement and relaxation. Normalized miRFP-paxillin and mCherry-DLC1 from 2 FAs shown in panels **A–C**. (**E**) Model of Rho GTPase activity modulation by DLC1 at FAs and at the plasma membrane relevant to *Figure 5*. Left and right panels show schematics of Rho activation dynamics in response to optoLARG optogenetic input at FAs (left panel) and plasma membrane (right panels). Top and bottom panels show schematics for Rho activation dynamics in response to optoLARG optogenetic input in control (top) and DLC1 KO (bottom) panels. (**F**) Model of force-dependent regulation of DLC1 at FAs relevant to this and *Figure 7—figure supplement 2*. Left panel, in the absence of acute mechanical input, DLC1 increases with FA assembly and decreases with FA disassembly. Central panel, upon acute local increase of mechanical stress in response to application of an optoLARG optogenetic input, DLC1 unbinds from FA in a reinforcement regime and rebinds FA in a relaxing regime when the optoLARG input is removed. Right panel, upon acute local increase of mechanical stress in response to application of an optoLARG optogenetic input, some FAs rupture after DLC1 dissociation and FA rupture.

The online version of this article includes the following figure supplement(s) for figure 7:

**Figure supplement 1.** DLC1 dynamics in an optoLARG stimulated FA that undergoes reinforcement followed by disassembly, as well as FA behavior in absence of optoLARG stimulus.

**Figure supplement 2.** DLC1 dynamics at FAs in unperturbed cells.

study the dynamic regulation of Rho by DLC1 at FAs because long-term loss of DLC1 function results in cells being locked in a state of excessive contractility.

To address this question, we used optogenetic control of Rho activation to probe different aspects of DLC1-mediated regulation of Rho activity at FAs, sites where DLC1 is highly enriched (*Figure 7*), and regulates Rho through talin- and tensin-dependent interactions, as well as at control regions on the PM, where DLC1 can bind lipids but is less abundant (*Erlmann et al., 2009*; *Haining et al., 2018*). In a first experimental modality, we evaluated the Rho activity flux at FAs as well as the PM in a mechanical regime that does not involve FA reinforcement (*Figure 5*). In the second experimental modality, we evaluated force-dependent DLC1-FA interaction in a mechanical regime in which FA reinforcement is clearly occurring (*Figure 5*).

In the first experimental modality, we probed the Rho GTPase flux under conditions that did not trigger FA reinforcement (*Figure 5D*). The goal was to probe DLC1-dependent regulation of Rho in the absence of strong mechanical strain. To achieve this, we applied small light inputs that induced only subtle fluctuations in Rho activity. Due to the weak signal-to-noise ratio (SNR) of these subtle GEF inputs, meaningful measurements of Rho activity response dynamics could only be obtained by averaging signals across hundreds of FAs. We observed that a transient synthetic optoLARG input applied to a FA yielded a Rho activation transient indicating the presence of tonic Rho GAP activity in the cell (*Figure 5E and F*). Interestingly, we observed that the identical synthetic optoLARG input applied to the PM in the absence of FA led to a Rho activity transient of lower amplitude. This suggests that FAs are intrinsically more permissive for GEF-mediated Rho activation, potentially due to local Src-dependent release of Rho from RhoGDI (*DerMardirossian et al., 2006*).

Since Rho activity may be regulated by DLC1 at FAs, we initially expected to observe altered Rho dynamics at these sites in *DLC1* KO cells, specifically that transient optoLARG input would lead to prolonged Rho activity. We found, however, that RhoA activation occurred faster, reached a higher peak amplitude, and returned to baseline more rapidly in both FA and non-FA PM regions compared to control cells (*Figure 5E and F*). Given that DLC1 is strongly enriched at FAs relative to the PM, the observation that similar RhoA flux changes occur at both subcellular locations suggests that these effects do not arise solely from local DLC1 activity. Rather, they may also reflect compensatory expression or activation of other GEFs and GAPs triggered by DLC1 loss. This interpretation is consistent with our experimental observation that *DLC1* KO cells engage compensatory mechanisms over time, resulting in weaker phenotypic differences than those observed in acute knockdown (*Figure 1*). However, such compensation likely lacks spatial specificity, as it does not adapt to local RhoA concentrations. This behavior mirrors findings in T cells, where Cdc42 deletion triggered compensatory transcriptional programs that restored cellular function (*Rochussen et al., 2025*), illustrating how Rho GTPase networks can rapidly adapt to gene loss and thereby mask the impact of individual regulators.

Our mechanistic model of dynamic RhoA modulation, fitted to experimental measurements of GEF input and resulting RhoA activity, further supports this view. The model revealed qualitative differences between WT and *DLC1* KO cells: WT cells exhibited Rho-dependent negative feedback that dampens activation dynamics, whereas this feedback was lost in the KO condition. The absence of this negative autoregulation explains the higher Rho activation amplitude observed in KO cells and implies that,

in the absence of DLC1, alternative Rho-regulated GAPs or regulatory proteins act independently of RhoA levels. Such negative feedback architectures are a hallmark of small GTPase signaling networks (*Bement et al., 2024*; *Tsyganov et al., 2012*; *Wu and Lew, 2013*), where they confer robustness and dynamic stability to signaling outputs. Altogether, our results uncover an unexpected complexity in the spatiotemporal regulation of Rho GTPase signaling at FAs, likely involving multiple GEFs and GAPs. They also highlight how long-term genetic perturbations can activate compensatory mechanisms, complicating the interpretation of single-gene KOs. Future experiments will require high-resolution, dynamic measurements of multiple GEFs, GAPs, and RhoA activity in unperturbed cells, or alternatively, acute perturbations that precede the onset of compensatory responses.

In the second experimental modality, we investigated mechanosensitive interactions between FAs and DLC1 under optoLARG stimulation conditions that promote robust FA reinforcement. We transiently increased actomyosin contractility within a small ROI, which triggered FA reinforcement followed by relaxation (*Figure 7*). Under such acute mechanical perturbations, we observed distinct DLC1 dynamics at FAs, likely involving its known mechanosensitive interactions with talin (*Haining et al., 2018*). In response to the induced strain, DLC1 rapidly dissociated from FAs exhibiting strong reinforcement (*Figure 7*; schematic in *Figure 7F*). This behavior correlates with in vitro findings showing that DLC1 dissociates from stretched talin due to unfolding of its R7-R8 domain (*Haining et al., 2018*). We also observed that FA relaxation promotes re-binding of DLC1 to FAs (*Figure 7*), suggesting that the R7-R8 talin domain might refold in these conditions. In this context, strong local mechanical input leading to FA reinforcement and talin stretching may cause rapid unbinding of DLC1 from FAs, transiently reducing local RhoGAP activity and providing a positive feedback loop that increases RhoA activation and local contractility. This may, in turn, promote FA disassembly as observed experimentally. Conversely, when the mechanical input ceases, DLC1 re-binding to relaxing FAs may rapidly suppress RhoA activity, restoring a low-contractility steady state. These nonlinear feedback mechanisms may enable amplification of local mechanical inputs to regulate all-or-nothing cytoskeletal events such as tail retraction during migration. Tail retraction is characterized by brief bursts of localized RhoA activity followed by rapid relaxation once retraction is complete (*Martin et al., 2014*; *Pertz et al., 2006*). In this model, SF pulling on FAs at the cell rear could trigger local DLC1 release, transiently elevate RhoA activity, and drive strong localized contraction leading to tail retraction. Testing this hypothesis experimentally is challenging, as long-term DLC1 perturbation will induce global hypercontractility, preventing cell polarization, directional migration, and thus the production of tail retraction events.

Summarizing the results from these two experimental regimes, we propose that DLC1 provides two distinct levels of Rho regulation. Under low mechanical input, DLC1 may help control global RhoA activity by positioning localized hotspots of RhoGAP activity throughout the cell, most likely together with other Rho GAPs. In contrast, under strong mechanical input leading to FA reinforcement, the DLC1-talin mechanosensitive interaction may permit rapid, local positive feedback on RhoA activity through DLC1 dissociation from FAs. This positive feedback could enable efficient FA turnover in response to mechanical stimuli and regulate processes such as tail retraction.

Our results illustrate how spatially controlled optogenetics with optoLARG enables spatiotemporal, reversible control of contractility, revealing mechanosensitive signaling interactions that are otherwise obscured by classical global perturbations of RhoA, such as nocodazole-induced microtubule depolymerization (*Krendel et al., 2002*) or lysophosphatidic-acid-mediated RhoA activation (*Pertz et al., 2006*) which trigger broad and nonspecific contractile responses. These findings provide new insight into the network circuitry governing spatiotemporal Rho GTPase signaling and point toward a new experimental modality for dissecting these mechanisms. By combining precise, reversible, and spatially confined optogenetic perturbations with automated optogenetic targeting, we can now locally modulate Rho activity and probe signaling fluxes across different regimes, using gentle perturbations to nudge the system out of equilibrium and reveal its relaxation back to steady state, or stronger inputs to engage mechanical feedback and adhesion reinforcement. Although this study focused on a GAP localized to FAs, the same approach can be extended to other cytoskeletal structures by targeting actin, myosin, or microtubule markers to guide light stimulation in real time. Such strategies will allow systematic mapping of how localized biochemical and mechanical feedback, mediated by multiple GEFs and GAPs, combine to generate emergent signaling patterns. Future studies systematically perturbing individual and combined GEFs and GAPs under controlled

optogenetic stimulation will be essential to identify their specific contributions to Rho GTPase flux and feedback architecture. Ultimately, achieving this level of quantitative, structure-specific control will be key to understanding how dynamic Rho GTPase signaling under the control of multiple GEFs and GAPs orchestrates cell morphogenesis.

# Materials and methods

## Cell lines

Rat Embryo Fibroblast 52 (REF52), a kind gift from Alan Howe, was cultured in Dulbecco's Modified Eagle Medium with 4.5 g/l glucose, 10% 4 mM L-Glutamine, and 100 U/ml penicillin/streptomycin. Cells were grown at 37 °C and 5% $CO_2$. Cells were regularly tested for mycoplasma contamination.

## siRNA knockdown

4 μl lipofectamine RNAi max was mixed with a total of 500 μl of OPTIMEM medium and a pool of siRNAs against the gene of interest at a final concentration of 60 nM (SiTools Biotech). The mix was added to a six-well plate containing 100,000 cells in a 2 ml medium volume medium.

## CrispR/Cas9 knockout generation

Two guide sequences targeting DNA within exon 5 of the DLC1 gene were selected using the CRISPOR tool for predicted high-specificity protospacer adjacent motif target sites in the rat genome (*Concordet and Haeussler, 2018*). Two complementary oligos each containing the DLC1 guide sequence and BbsI ligation adapters were synthesized (Microsynth; sense sequence: 5'-CACCGAAC CGAGAGAGCTACCCGG-3', antisense sequence: 5'-AAACCCGGGTAGCTCTCTCGGTTC-3'). The guide sequences were annealed and ligated into a pSpCas9(BB)–2A-GFP vector. REF52 cells were grown in six-well plates to 60% confluency and transfected with 1 μg of the vector together with 1 μl of Lipofectamine 3000 and 5 μl of P3000 solution (Thermo Fisher Scientific). Two days post-transfection cells were detached, suspended in PBS +1% FBS, and sorted into 96-well plates using fluorescence-activated cell sorting. After expansion, individual clones were detached and suspended in the lysis buffer (10 mM TRIS, 50 mM KCL, 2.5 mM MgCl2, 0.45% Tween-20, 0.05% Gelatin, 0.12 mg/ml of proteinase K). The cell solution was frozen at –80 °C and subsequently kept at 63 °C for 1 hr. Proteinase K was inactivated by heating the solution to 95 °C for 15 min. PCR primers were designed to amplify a 500 bp region around the CRISPR cut site (forward primer: 5'-AAGGAGTGTGTCTAACTCCACGCA GACCAG-3', reverse primer: 5'-CTCCTTAGGACTGTCGCTGCTGTTTTCTCT-3'). Genomic sequences were amplified by PCR and sequenced by Sanger sequencing.

## DNA constructs

A 'pB3.0' piggybac vector was created by adapting the pPBbsr2 (*Matasci et al., 2011*) to make it smaller in size. pPBbsr2 was digested with AscI and PacI, yielding 3922 bp and 2864 bp fragments. Full-size pPBbsr2 was used as template for a PCR with forward primer 5'-TTAGCATTAATTAAGCG GCCGCGTTGCTGGCGTTTTTCC-3' and reverse primer 5'-GTGCCTTTACAACTTATGAGTAACCCCG CGCGGACGATT-3', yielding a 1671 bp insert that was ligated back into the receiving 3922 bp part of the digested pPBbsr2 vector, creating 'pB3.0-BLAST' (Blasticidine resistance). To introduce different resistance cassettes, we performed an overlap extension PCR with pB3.0-BLAST as template for PCR1 (primers A+B) and PCR2 (primers C+D) with the following primers:

primer A: 5'-AAGGATGCCCAGAAGGTACCCCATTGTATGGGATCTGATCTGGG-3', primer B: 5'-GGAAACTTTTTGTGCTATTATGGTGGCCATTCAGCTCTACGTAGCTACT-3', primer C: 5'-CCTT TGAAAAACACGATAATACCACCGGTAAGTCGAGATGCATCGATGA-3', primer D: 5'-CTCCGCCT TTCTTGGACGTCGGGTTCGAACCGCATTAGT-3'. Primers A and D, the product of PCR1 and PCR2, were used for fusing the two fragments by PCR. The resulting PCR product was ligated into the pB3.0-BLAST vector, digested with Kpn1 and Pst1, by fusion cloning, yielding a pB3.0-noAB vector with a single Age1 site to introduce antibiotic resistance cassettes. To produce pB3.0-HYGRO and pB3.0-PURO, PCRs were performed on template vectors pHygro-PB and pPuro-PB, respectively. For pB3.0-HYGRO forward primer 5'-AACACGATAATACCACCATGAAAAAGCCTGAACTCACCGC-3' and reverse primer 5'-GCAGGCTCCCGTTTCCTTATCGGCCATTCAGCTCTA-3' were used. For pB3.0-PURO forward primer 5'-AACACGATAATACCACCATGACCGAGTACAAGCCCACG-3' and reverse

primer 5'-GCGTTCGGGCCACGGACTGGCCATTCAGCTCTA-3' were used. The respective inserts were inserted via fusion cloning in the pB3.0-noAB vector, digested with AgeI.

Construction of the pB3.0-optoLARG-mVenus-SspB-p2A-stargazin-mtq2-iLID optoLARG system was as follows. The cDNA for optoLARG-mVenus-SspB(nano)-p2A-iLID-CAAX was synthesized by custom gene synthesis (Genewiz, Azenta Life Sciences, US). The sequence contained the catalytical DH-PH domain of human RhoGEF12 (LARG, aa 766–1138), connected via a linker (GSGSGSGS) to full-length mVenus fused to SSPB(nano). This is followed by a p2A sequence (GSGATNFSLLKQ AGDVEENPGP), the iLID module (*Guntas et al., 2015*), and a CAAX box (KRAS). The construct was cloned into pB3.0-BLAST containing a CAG promoter, yielding pB3.0-optoLARG-mVenus-SspB-p2A-iLID-CAAX. After publication of an improved PM anchor for the iLID module (*Natwick and Collins, 2021*), we changed the part after the p2A sequence. The pB3.0-optoLARG-mVenus-SspB-p2A-iLID-CAAX vector was digested with BsrGI and AflII, and a custom gene synthesized stretch of cDNA containing SspB(nano)-p2A, full-length stargazin, mTurqoise2, and iLID was inserted. This yielded pB3.0-optoLARG-mVenus-SspB(nano)-p2A-stargazin-mtq2-iLID (exact sequences of the gene synthesis products are available on request). We also developed a version with the recruitable SSpB domain tagged in red (mScarlet3) instead of yellow (mVenus), which allowed us to image optoLARG recruitment without activating it.

An eGFP DLC1 plasmid containing the 1091 aa long isoform originating from mouse (kindly provided by Monilola A. Olayioye) was used to produce mCherry-DLC1. The DLC1 gene was cut out of this plasmid with BamHI and N-terminally fused to FKBP12-mCherry and inserted into pB3.0-BLAST containing a CAG-promoter, yielding pB3.0-FKBP12-mCherry-DLC1.

## Spreading assay

Glass-bottom well plates (Celvis) were coated with 5 µg/ml of human plasma fibronectin purified protein (Merck) for 1 hr at room temperature. REF52 cells were seeded at a density of 7000 cells per well (using 24-well plates) in FluoroBrite DMEM medium containing 1% FBS, 0.1% BSA, 4 mM L-Glutamine, and 100 U/ml penicillin/streptomycin. Imaging was done with a Nikon Eclipse Ti-E inverted microscope with an automatic stage. Temperature was kept at 37 °C with a temperature control system, humidity (100%), and $CO_2$ (~5%) with a gas mixer (Life Imaging Services). Focus drift was prevented by the equipped Perfect Focus System (Nikon). The microscope was controlled with Metamorph software (Universal Imaging).

## Optogenetic stimulation experiments

Glass-bottom well plates (Celvis) were coated with 5 µg/ml of human plasma fibronectin purified protein (Merck) for 1 hr at room temperature. REF52 cells were seeded at 6000 cells/well into 12-well plates and allowed to attach for 12 hr. For both RhoA activity and mCherry-DLC1/miRFP-paxillin optogenetic stimulation experiments, cells were incubated in FluoroBrite DMEM medium overnight. The microscope was controlled with NIS-Elements (Nikon) for observing DLC1 dynamics and with open-source micromanager software (*Edelstein et al., 2014*) for edge stimulation. Optogenetic stimulation was localized to specific regions using an Andor mosaic 3 DMD. For mCherry-DLC1/miRFP-paxillin experiments, ROIs were user-defined. For RhoA activity experiments, cells in different fields of view were selected by a user, cells were then segmented with a custom pixel-classifier based on a pre-trained VGG16 convolutional neural network (*Hinderling et al., 2026*), which was trained on a field of view of the experimental data. ROIs for stimulation masks were then automatically defined at the top and bottom of the cell for consecutive illumination with different light pulse regimes using our method for automated optogenetic targeting (*Hinderling et al., 2025*). Fields of view were imaged one after the other on multiple cells. For data extraction, the stimulation mask was expanded by Gaussian smoothing and then thresholded to cover a wider region, reducing artifacts caused by membrane dynamics. Image intensities were calculated for the overlap of this mask with the segmented cells and the full segmented cells, respectively.

## Immunohistochemistry

One hour after seeding, cells were fixed for 10 min with 0.2% paraformaldehyde. A 0.1% Triton X solution was used for permeabilization, after which cells were incubated overnight at 4 °C with the blocking buffer, containing 5% Bovine-Serum-Albumin and 0.05% Tween20 in PBS. After each step

of the following protocol, cells were washed three times with Phosphate-Buffered-Saline (PBS). Cells were incubated overnight with the primary antibody 1:250 dilution for paxillin (Abcam ab32084), 1:50 for pMLC (Cell Signaling Antibody #3671). Secondary antibody incubation (1:1000) lasted 1 hr. Phalloidin incubation (1:200 Phalloidin-Atto647N) was performed for 20 min and DAPI incubation for 10 hr. Cells were imaged immediately after the last step.

## Image analysis

Lamella size, FA areas (*Figure 1*), and RhoA activity ratio measurements (*Figure 3*) were computed with CellProfiler in conjunction with a pixel classifier trained with Ilastik (*Berg et al., 2019*). Spreading dynamics (*Figure 2D*) were manually analyzed. FRET ratio images were computed and analyzed with custom python scripts. DLC1 and paxillin dynamics (Figure S4D) were calculated with manually drawn ROIs in Fiji (*Schindelin et al., 2012*).

## ODE model and data fitting

For data fitting and parameters optimization, we used the Julia programming language and tools made available by the Scientific Machine Learning (SciML) organization as framework. The packages ModelingToolkit.jl (*Ma et al., 2022*) and DifferentialEquations.jl (*Rackauckas and Nie, 2017*) allowed us to symbolically define and solve the ODE system. The ODE solver algorithm was specified as the Rodas4P method. In addition, we used the automatic switching algorithm AutoVern9 (*Verner, 2010*) for stiffness detection. For the parameter optimization, we defined the problem with the package Optimization.jl (*Dixit and Rackauckas, 2023*). The gradient-based method L-BFGS (*Nocedal and Wright, 2006*) was used as optimization solver from the packages Optim.jl (*K Mogensen and N Riseth, 2018*). Additionally, the packages SciMLSensitivity.jl (*Rackauckas et al., 2021*) and Forward-Diff.jl (*Revels et al., 2016*) were required for an efficient and automatic computation of the gradient with forward propagation. Initial conditions were set to zero for all model species ($\text{Rho}_0^*$, $\text{GAP}_0^*$, $\text{GEF}_0^*$). Total protein concentrations were assumed to be conserved between active and inactive forms. Given the low concentrations at which the system operates, and the relatively small fraction of protein that becomes activated, changes in the inactive pool were considered negligible. As a result, the inactive species were assumed to remain constant throughout the simulation.

To estimate the activation and deactivation rates of optoLARG, we fitted a simple linear model to experimental measurements of membrane recruitment. In the case of the RhoA signaling model, Michaelis-Menten kinetics were used to model the interactions between active species, apart from GAP inactivation, which was treated as a first-order (linear) deactivation process.

In the *DLC1* KO condition, the Michaelis-Menten constant (*Km*) of RhoA-dependent GAP activation consistently converged toward zero during optimization, indicating a loss of negative feedback regulation. To reflect this and simplify the model, we explicitly set this parameter to zero, resulting in a reduced model in which GAP activity is independent of RhoA levels. This simplified configuration still produced a strong fit to the experimental data, supporting the hypothesis that the feedback loop regulating GAP activity is disrupted in the KO condition.

We also assessed whether model parameters are uniquely recoverable in principle from the observation scheme using StructuralIdentifiability.jl (*Dong et al., 2023*). For both the WT and *DLC1* KO ODE model presentations, we ran the package's known-initial-conditions workflow with our initial states treated as known and fixed (here set to 0) and default probabilistic settings (probability threshold 0.99). The analysis returns whether each parameter (and selected parameter functions) is globally identifiable given the model equations and measured outputs, that is independent of data noise or sampling.

The following table summarizes the results of the structural identifiability analysis:

| Parameter | Membrane (WT) | Membrane (*DLC1* KO) | FAs (WT) | FAs (*DLC1* KO) | Comment |
|---|---|---|---|---|---|
| $V_{\text{ON}}^{\text{GAP}}$ | 25.2255 | 21.51569 | 33.706 | 15.6152 | *alias $k_{\text{ON}}^{\text{GAP}}$ for the DLC1 KO adapted model* |
| $Km^{\text{GAP}}$ | 0.093982 | →0 | 0.205939 | →0 | *consistent across fits* |

*Continued on next page*

*Continued*

| Parameter | Membrane (WT) | Membrane (*DLC1* KO) | FAs (WT) | FAs (*DLC1* KO) | Comment |
|---|---|---|---|---|---|
| $k_{\mathrm{OFF}}^{\mathrm{GAP}}$ | 66.0697 | 68.84138 | 99.9969 | 59.1818 | *varied with upper parameter bound* |
| $V_{\mathrm{ON}}^{\mathrm{Rho}}$ | 0.0061944 | 0.00997069 | 0.00457155 | 0.00759578 | *consistent across fits* |
| $Km_{\mathrm{ON}}^{\mathrm{Rho}}$ | 0.0118618 | 0.031674 | 0.0108763 | 0.0382543 | *consistent across fits* |
| $V_{\mathrm{OFF}}^{\mathrm{Rho}}$ | 25.2258 | 21.5156 | 26.7483 | 15.6151 | *varied with upper parameter bound* |
| $Km_{\mathrm{OFF}}^{\mathrm{Rho}}$ | 60.6878 | 94.83013 | 76.8346 | 98.2568 | *varied with upper parameter bound* |

## Code availability

The code used to obtain the presented results can be found in the GitHub repository *RhoAModelling* (https://github.com/girochat/RhoAModelling; copy archived at *Rochat, 2025*) as Pluto and Jupyter notebooks. It contains the model and data fitting of RhoA dynamics at FAs and membrane as well as the parameter estimation of optoLARG dynamics. The Pluto.jl package offers a Julia programming environment analog to Jupyter notebook. For reproducibility, the repository describes how to set a similar coding environment and references all packages that were used.

## Acknowledgements

This work was supported by the Swiss National Science Foundation IZSAZ3_173462 Argentinian-Swiss Joint Research Programme and by the Sinergia CRSII5_183550 grants to Olivier Pertz. We are grateful to the Microscopy Imaging Center of the University of Bern for support (https://www.mic.unibe.ch). We thank Kazuhiro Aoki (NIBB, Japan) for providing the pPBbsr2 vector, David L Hacker (EPFL, Switzerland), for the pPuro-PB and pHygro-PB plasmids, Judith Trüb (UniBE, Switzerland) for the cloning of pB3.0-BLAST, Monilola A Olayioye (University of Stuttgart, Germany) for the eGFP DLC1 plasmid, Dean E Natwick and Sean R Collins (University of California Davis) for sharing the Stargazin-iLID constructs. We are grateful to Miguel Vicente-Manzanares, Bernhard Wehrle-Haller, and Daniel Riveline for constructive comments on the manuscript.

## Additional information

### Funding

| Funder | Grant reference number | Author |
|---|---|---|
| Schweizerischer Nationalfonds zur Förderung der Wissenschaftlichen Forschung | IZSAZ3_173462 | Olivier Pertz |
| Schweizerischer Nationalfonds zur Förderung der Wissenschaftlichen Forschung | CRSII5_183550 | Olivier Pertz |

The funders had no role in study design, data collection and interpretation, or the decision to submit the work for publication.

### Author contributions

Lucien Hinderling, Conceptualization, Software, Investigation, Visualization, Methodology, Writing – review and editing; Max Heydasch, Conceptualization, Validation, Investigation, Visualization, Methodology, Writing – original draft; Giliane Rochat, Software, Formal analysis, Visualization, Methodology;

Laurent Dubied, Jakobus van Unen, Investigation, Methodology; Maciej Dobrzynski, Conceptualization, Software, Methodology; Olivier Pertz, Conceptualization, Supervision, Funding acquisition, Writing – original draft, Writing – review and editing

### Author ORCIDs
Lucien Hinderling https://orcid.org/0000-0002-3956-9363
Laurent Dubied https://orcid.org/0009-0008-7772-330X
Maciej Dobrzynski https://orcid.org/0000-0002-0208-7758
Olivier Pertz https://orcid.org/0000-0001-8579-4919

Reviewer #1 (Public review): https://doi.org/10.7554/eLife.90305.3.sa1
Author response https://doi.org/10.7554/eLife.90305.3.sa2

---

# Additional files

### Supplementary files
MDAR checklist

### Data availability
Code and data is available from https://github.com/girochat/RhoAModelling/ (copy archived at *Rochat, 2025*).

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
