## [Editor Report · eLife Assessment]

This study presents a **valuable** finding on how the GAP DLC1, a deactivator of the small GTPase RhoA, regulates RhoA activity globally as well as at Focal Adhesions. Using a new acute optogenetic system coupled to a RhoA activity biosensor, the authors present **convincing** evidence that DLC1 amplifies local Rho activity at Focal Adhesions. Thanks to modeling, they show that DLC1 is needed for a negative feedback loop that engage more RhoA deactivators upon RhoA activation, highlighting the complex regulation of RhoGTPases in space and time.

---

## [Referee Report · Reviewer #1 (Public review)]

Summary:

The manuscript of Heydasch et al. addresses the spatiotemporal regulation of Rho GTPase signaling in living cells and its coupling to the mechanical state of the cell. They focus on a GAP of RhoA, the Rho specific GAP Deleted in Liver Cancer 1 (DLC1). They first show that removing DLC1 either by a CRISPR KO or by downregulation using siRNA leads to an increased contractility and globally elevated RhoA activity, as revealed by a FRET biosensor. This result was expected, since DLC1 is deactivating RhoA its absence should lead to increasing amounts of active RhoA. To go beyond global and steady levels of RhoA activity, the authors developed an acute optogenetic system to study transient RhoA activity dynamics in different genetic and subcellular contexts. In WT cells, they found that pulses of activation lead to an increased RhoA activity at focal adhesions (FA) compared to plasma membrane (PM), which suggests that FAs contain less RhoA GAPs, more RhoA, or that FAs involve positive feedbacks implying others GEFs for example. In DLC1 KO cells, they found that the RhoA response upon pulses of optogenetic activation was increased (higher peak) both at FA and PM, which could be expected since less GAP should increase the amount of active RhoA. But surprisingly, they observed also a higher rate of RhoA deactivation in DLC1 KO cells, which is counterintuitive: less GAP should result in a slower rate of deactivation. Less GAP should also lead to a lower rate of observed RhoA activation (smaller koff) and delayed peak. Using a modeling approach and control experiments (to monitor the optogenetic intrinsic dynamics), the authors propose that there is a negative feedback in WT cell between activated RhoA and the activity of its GAPs (other than DLC1). More active RhoA decreases GAP activity such that active RhoA relaxation to its basal state is relatively slow. This negative feedback would be absent in DCL1-deficient cells, explaining the relatively faster relaxation. This hypothesis is convincing given the data and the model, and it shows that there are compensatory mechanisms at play when DLC1 is knocked down. Further on, the authors study the dynamics of DLC1 on FAs depending on the mechanical state and nicely show a causal decrease of DLC1 enrichment at FA upon FA reinforcement, hereby probing a positive feedback where RhoA activation is further amplified as the force exerted at FA is increasing. Altogether, this work highlight the extremely fine regulation in space and time of RhoGTPases that is only revealed through acute perturbations, while at the cell scale and long time scale, complex compensatory mechanisms are at play rendering knock-down or overexpression experiments not always straightforward to interpret (in the present case, knock-down of a deactivator lead to an increase of deactivation rate through the induced absence of other activity dependent-deactivators).

Strengths:

- Experiments are precise and well done.

- Technically, the work brings original and interesting data. The use of transient optogenetic activation within focal adhesions together with a biosensor of activity is new and elegant.

- The link between DLC1 and global contractility/RhoA activity is clear and convincing.

- The surprising higher rate of RhoA deactivation in DLC1 KO cells is convincing, as well as the differences in the dynamics of RhoA between focal adhesions and plasma membrane.

- The model is very helpful to support the hypothesis of the negative feedback loop.

- The correlation between DLC1 enrichment and focal adhesion dynamics is very clear.

Weaknesses:

- The negative and positive feedback loops could have been dug more deeply molecularly (in particular discover what are the compensatory mechanisms at play), but this could be the purpose of future work.

Comments on revised version:

I thank the authors for the great improvement of their work and their detailed answers to my comments. The modeling work is great and really brings novelty to the story. It also helps a lot to have the data for the optoLARG recruitment. I suggest authors move to the Version of Record.

---

## [Author Response]

The following is the authors’ response to the original reviews.

We thank the reviewers for their careful reading of our manuscript and for the constructive and insightful feedback. In response, we performed several new experiments and analyses that significantly strengthen the study. First, we addressed the important question of optoLARG recruitment dynamics by generating a new cell line expressing optoLARG-mScarlet3 together with paxillin-miRFP, enabling us to directly quantify the dynamics of the optogenetic activator at focal adhesions and the plasma membrane. Second, we introduced a quantitative modeling framework to analyze RhoA activity dynamics during transient optogenetic stimulation. Using the measured optoLARG kinetics as input, we fitted activation and deactivation parameters for both WT and *DLC1* KO cells, revealing a loss of negative feedback regulation in the KO condition. Together, these additions clarify the temporal relationships between optogenetic activation, RhoA signaling, and biosensor responses, and provide a more rigorous, mechanistic interpretation of our data. We rewrote large parts of the discussion section to reflect this new information.

Below, we provide detailed, point-by-point responses to all reviewer comments.

**Recruitment dynamics optoLARG**

**Reviewer #1:**

**Public Review:**
For the optogenetic experiments, it is not clear if we are looking at the actual RhoA dynamics of the activity or at the dynamics of the optogenetic tool itself.
**Recommendations for the authors:**
For the transient optogenetic activations at FA and PM, it would be great to have one data set where the optoLARG is fused to a fluorescent protein, for example, mCherry, while FAs would be marked with paxillin-miRFP (by transient transfection to avoid making a new stable cell line). The dynamics of the optogenetic activator should be the same (on and off rates), but it can be possible that the activator is retained at FA for example. Such an experiment would help the understanding of the differential observed dynamics, where several timescales are involved: the dynamics of the opto tool, the dynamics of RhoA itself, and the dynamics of the biosensor.

We agree with the reviewers, this is an essential control for this manuscript and the cell line will be useful in future studies. We developed a new construct containing with the recruitable SSpB domain tagged in red (optoLARG-mScarlet3) compatible with the iLid system, and paxilin-miRFP to locate the focal adhesions. From previous experiments we know that the anchor part of optoLARG system is distributed evenly across the cell membrane and is not affected by cytoskeletal structures like focal adhesions. As for the recruitable part of the optoLARG system, that translocates from the cytosol to the membrane upon blue light stimulation, we illuminated focal adhesion and non-focal adhesion regions, and quantified optoLARG dynamics. The same scripts were used for automated stimulation and analysis as were used for the rGBD recruitment experiments. We illustrate these results in the new Suppl. Fig S3. We found no significant difference in recruitment dynamics between focal adhesion/non-focal adhesion regions (Fig. S3B). We found the optoLARG dynamics fits well with inverse-exponential during recruitment under blue light stimulation, and exponential decay after blue light stimulation (disassociation phase), consistent with the expected iLID dynamics (Fig S3C). This experiment is described in detail at the end of the section "Optogenetic interrogation of the Rho GTPase flux in WT and DLC1 KO cells" (Lines 303-320). We then went on to use the optoLARG dynamics as input for the models describing RhoA activity dynamics (see next comment). This should help to untangle the measured RhoA dynamics from the dynamics of the optogenetic tool.

**Quantitative analysis RhoA activity dynamics**

**Public Review:**
There is no model to analyze transient RhoA responses, however, the quantitative nature of the data calls for it. Even a simple model with linear activation-deactivation kinetics fitted on the data would be of benefit for the conclusions on the observed rates and absolute amounts.
**Recommendations for the authors:**
[...] for the transient optogenetic experiments, it would be great to make a simple model, or at least to fit the curves with an on rate, an off rate, and a peak value. This will clarify the conclusions drawn for the experiments. For example, the authors claim that they observe an increased Rho activation rate in DLC1 KO cells (see sections "Optogenetic interrogation of the Rho GTPase flux in WT and DLC1 KO cells" and "Discussion") but the rate is not well-defined. One can have two curves with the same activation rate but one that peaks higher (larger multiplicative prefactor) and it would resemble the presented data. This being said, the higher deactivation rate in DLC1 KO cells is evident from the data.

We agree that a quantitative analysis and model would improve our understanding of the data. We fit the activation/deactivation kinetics and provide the values in the chapter "Optogenetic interrogation of the Rho GTPase flux in WT and DLC1 KO cells" (Lines 287-299). We then modeled the RhoA activity dynamics at focal adhesions and at the plasma membrane after transient optogenetic stimulation using a system of ODEs, using the new measurements of optoLARG kinetics as activation input. We find a close fit for the experimental data, with WT following classic Michaelis-Menten dynamics. Interestingly, when fitting the DLC1-KO data with the same model as for WT, the parameter modeling the negative feedback loop (active RhoA recruiting a GAP) is set to zero; in other words, the factor that deactivates RhoA is present at a constant concentration. We added an additional main Figure 5 describing the models and fits, and added a new Results section "Modeling indicates loss of negative RhoA autoregulation in DLC1-KO cells" (Lines 326-378), and also updated the Methods and Discussion section of the paper accordingly. We use the findings to more clearly ground the mathematical terms used to describe our results.

**Error figure 6E**

**Recommendations for the authors:**
The scheme presented in Figure 6E is not supported by the data and should be modified. In this scheme, the authors show a strongly delayed peak in control cells versus DCL1 KO cells, whereas in the data the peaks appear to be at similar time points. Similarly, the authors show a strongly decreased rate of activation, whereas the initial rates appear identical in the data.

The delayed peak we illustrated is an error, we thank the reviewers for catching it. The decreased rate of deactivation and activation, although exaggerated in the scheme, is however present in the data (and is now quantified, see answer above). We updated the figure accordingly (now Fig. 7E in the manuscript).

**Clarification term "signaling flux"**

**Recommendations for the authors:**
It would be nice to define more precisely several terms that are used throughout the manuscript. For example, could the authors define what they mean by "signaling flux"? Is it the temporal derivative of the Rho levels? Or the spatial derivative?

We agree that this was not clear in the previous version of the manuscript. We refer to "signaling flux" as the continuous cycle of RhoA activation by GEFs and inactivation by GAPs, processes that persist even when bulk RhoA activity appears steady, as introduced by Miller & Bement (2009). We now explicitly define "signaling flux" in the abstract (Lines 20-24).

See: Miller, Ann L., and William M. Bement. "Regulation of cytokinesis by Rho GTPase flux." Nature cell biology 11.1 (2009): 71-77. https://doi.org/10.1038/ncb1814

**Recommendations for the authors:**
Also (see above) it would be nice to define precisely what are the rates: the activation rate is in general the k_on of a reaction scheme, but it will differ from the observed rate given by a biosensor. For example, with a k_on and a k_off the observed rate toward the steady-state will be given by the sum of the activation and deactivation rates. In the manuscript, the authors do not make the distinction between the activation rate with the rate of increase of the biosensor which is confounding for the reader and for the interpretation of the data.

We update the results section to make this distinction more clear (Lines 288-300), and add a note explicitly highlighting the difference between biosensor signal dynamics and the underlying RhoA activation/deactivation rates (Lines 298-300). In addition, our newly introduced model helps disentangle the combined activation/deactivation rates into distinct GEF and GAP activity parameters.

**Improvements to figure 3**
Minor recommendation:In Figures 3 B and D, the stars (statistical differences) are not visible. It would be good to make them bigger or move them above the graphs.

Thank you! We updated the graphics.

Other changes

Additional panel (Figure 5D) showing paxillin intensity does not change after weak optogenetic stimulation, to better illustrate the weak stimulation regime that does not trigger FA reinforcement (contrasting Figure 7). Additional small layout changes to Figure 5.

Addition of authors that contributed to the revisions